# Chemotactic network responses to live bacteria show independence of phagocytosis from chemoreceptor sensing

## Netra Pal Meena*, Alan R Kimmel*

Laboratory of Cellular and Developmental Biology, National Institute of Diabetes and Digestive and Kidney Diseases, The National Institutes of Health, Bethesda, United States

**Abstract** Aspects of innate immunity derive from characteristics inherent to phagocytes, including chemotaxis toward and engulfment of unicellular organisms or cell debris. Ligand chemotaxis has been biochemically investigated using mammalian and model systems, but precision of chemotaxis towards ligands being actively secreted by live bacteria is not well studied, nor has there been systematic analyses of interrelationships between chemotaxis and phagocytosis. The genetic/molecular model *Dictyostelium* and mammalian phagocytes share mechanistic pathways for chemotaxis and phagocytosis; *Dictyostelium* chemotax toward bacteria and phagocytose them as food sources. We quantified *Dictyostelium* chemotaxis towards live gram positive and gram negative bacteria and demonstrate high sensitivity to multiple bacterially-secreted chemoattractants. Additive/competitive assays indicate that intracellular signaling-networks for multiple ligands utilize independent upstream adaptive mechanisms, but common downstream targets, thus amplifying detection at low signal propagation, but strengthening discrimination of multiple inputs. Finally, analyses of signaling-networks for chemotaxis and phagocytosis indicate that chemoattractant receptor-signaling is not essential for bacterial phagocytosis.

**\*For correspondence:** meenan@ mail.nih.gov (NPM); alank@helix. nih.gov (ARK)

**Competing interests:** The authors declare that no competing interests exist.

## Introduction

Professional phagocytes hunt and destroy bacteria and other small non-related single-celled organisms, and survey for apoptotic or necrotic cells and cell remnants (*Boulais et al., 2010*; *Freeman and Grinstein, 2014*; *Niedergang et al., 2016*). While these functions originally served to fulfill nutritional requirements (*Cosson and Soldati, 2008*; *de Nooijer et al., 2009*), they have been re-purposed as a first line innate immune response in the complex metazoan. Still, phagocytes that have either retained ancient utility for foraging (*e.g. Dictyostelium*) or display defensive or developmental postures (*e.g.* macrophages) share common mechanistic pathways for chemotaxis (detection) and phagocytosis (killing), despite their evolutionary separation $\sim 10^3$ myo (*Bozzaro et al., 2008*; *Cosson and Soldati, 2008*; *Jin et al., 2008*; *de Nooijer et al., 2009*; *Bozzaro and Eichinger, 2011*; *Artemenko et al., 2014*).

Both *Dictyostelium* and mammalian cells utilize GPCRs (G protein-coupled receptors) for chemoattractant/chemokine detection and related downstream pathways for activation response (*e.g.* Ras, PKA, PI3K, TORC2, AKT, PDK1, ERKs, Cyclases, Actin (*Van Haastert and Veltman, 2007*; *Jin et al., 2008*; *McMains et al., 2008*; *Swaney et al., 2010*; *Jin, 2013*; *Artemenko et al., 2014*). Chemotaxis/migration of *Dictyostelium* and mammalian cells are similarly inhibited by latrunculin A, PD

169316, SB 525334, and other small molecules (*Liao et al., 2016*). Phagocytic processes are also highly conserved and mechanistically common to both *Dictyostelium* and macrophages (*Gotthardt et al., 2006*; *Bozzaro et al., 2008*; *Cosson and Soldati, 2008*; *Boulais et al., 2010*; *Freeman and Grinstein, 2014*; *Levin et al., 2016*; *Niedergang et al., 2016*). Particle interaction at the cell surface induces an actin re-organization, forming a cup in the plasma membrane that surrounds the particle for engulfment and re-direction to lysosomes for degradation. The shared mechanisms also underscore the great utility for model studies of host-pathogen interactions, where cells of the mammalian immune system and *Dictyostelium* exhibit similar susceptibility to virulence infection by *Legionella pneumophila*, *Mycobacterium tuberculosis*, and *Listeria monocytogenes* (*Bozzaro et al., 2008*; *Steinert, 2011*; *Levin et al., 2016*; *Gray and Botelho, 2017*).

Many platforms exist to image and assess chemotaxis in defined chemical gradients (*Ponath et al., 2000*; *Sawai et al., 2007*; *Kato et al., 2008*; *Hattori et al., 2010*; *Timm et al., 2013*; *Veltman et al., 2014*; *Collins et al., 2015*; *Guckenberger et al., 2015*; *Liao et al., 2016*; *Mackenzie et al., 2016*). Thus, it has been possible to identify GPCRs for specific chemoattractant ligands, intracellular pathways that become highly polarized in comparison to shallow extracellular gradients, and signaling networks that regulate directional cytoskeletal responses (*Artemenko et al., 2014*; *Graziano and Weiner, 2014*; *Nichols et al., 2015*). Still, quantitative analyses of definitive bacterial sensing have been limited. Here, we utilize growth-phase *Dictyostelium* to dissect differential responses to live gram positive and gram negative bacteria and, thereby, define high sensitivity to bacterial populations and their secreted chemoattractants (<0.5 nM) and conditions for additive or competitive responses to multiple chemoattractants. We show that multiple, low signal inputs provide a mode to amplify detection for chemotaxis to limited bacterial numbers. Conversely, since adaptive pathways for individual chemoattractants are mechanistically separated, cells can discern and migrate within a defined chemical gradient, even in the presence of other non-directional, saturating signals.

Finally, we considered if chemotactic cell surface receptor signaling *via* pseudopod extension mediates phagocytosis through locally enhanced host-bacterial interactions (*Swanson and Baer, 1995*; *Bozzaro et al., 2008*; *Heinrich and Lee, 2011*; *Levin et al., 2016*; *Niedergang et al., 2016*; *Gray and Botelho, 2017*). Utilizing strains lacking or expressing specific chemoattractant GPCRs, we demonstrate that phagocytosis is minimally dependent on bacterial chemoattractant sensing; GPCR-deficient lines exhibit similar high efficiencies for phagocytosis as do corresponding strains that express WT (wild-type) GPCRs, regardless of their ability to respond chemotactically.

## Results

### Chemotaxis of growth-phase *Dictyostelium* to folate

The Essen IncuCyte system allows real-time imaging of chemotaxis in 96 individual compartments of a multi-tiered, micro-well plate. Each of the 96 compartments is comprised of an upper and lower chamber, separated by a membrane, with uniformly-spaced 8 μm pores. Chemoattractants are added to the bottom well of the chambers, and ~$10^3$ *Dictyostelium* cells are allowed to adhere to the surface of the membrane at the base of the upper well. A chemoattractant gradient is established by chemical diffusion from the bottom well, through the membrane pores, and into the upper well; chemotactic cell migration is quantified over time, by imaging cells that had migrated from the upper well base, through the pores, to the under surface of the membrane within the bottom chamber. The assays are highly reproducible, with generally <10% non-specific directed migration to phosphate buffer controls (see *Figure 1A and B*, and *Videos 1* and *2*).

*Dictyostelium* grow as single cells that can migrate to pterin-derived folate moieties in a search for bacterial nutrient sources. *Figure 1B* illustrates a typical chemotactic dose response, time-course for migration of growth-phase *Dictyostelium* toward folate. Under these conditions, growing cells migrate toward folate with an EC$_{50}$ ~5 nM (*Figure 1C*). While we recognize that this is a calculated maximum for cell exposure to folate, it defines a very high detection/migration sensitivity (<0.5 nM) and emphasizes the robust nature for chemotaxis in this assay.

To evaluate specificity of the migration response, we analyzed WT and mutant cells with differential sensitivities to folate. Cells lacking G$\beta$, an essential subunit for all receptor/G protein signaling (*Wu et al., 1995*; *Peracino et al., 1998*; *Hoeller et al., 2016*) were completely unresponsive to

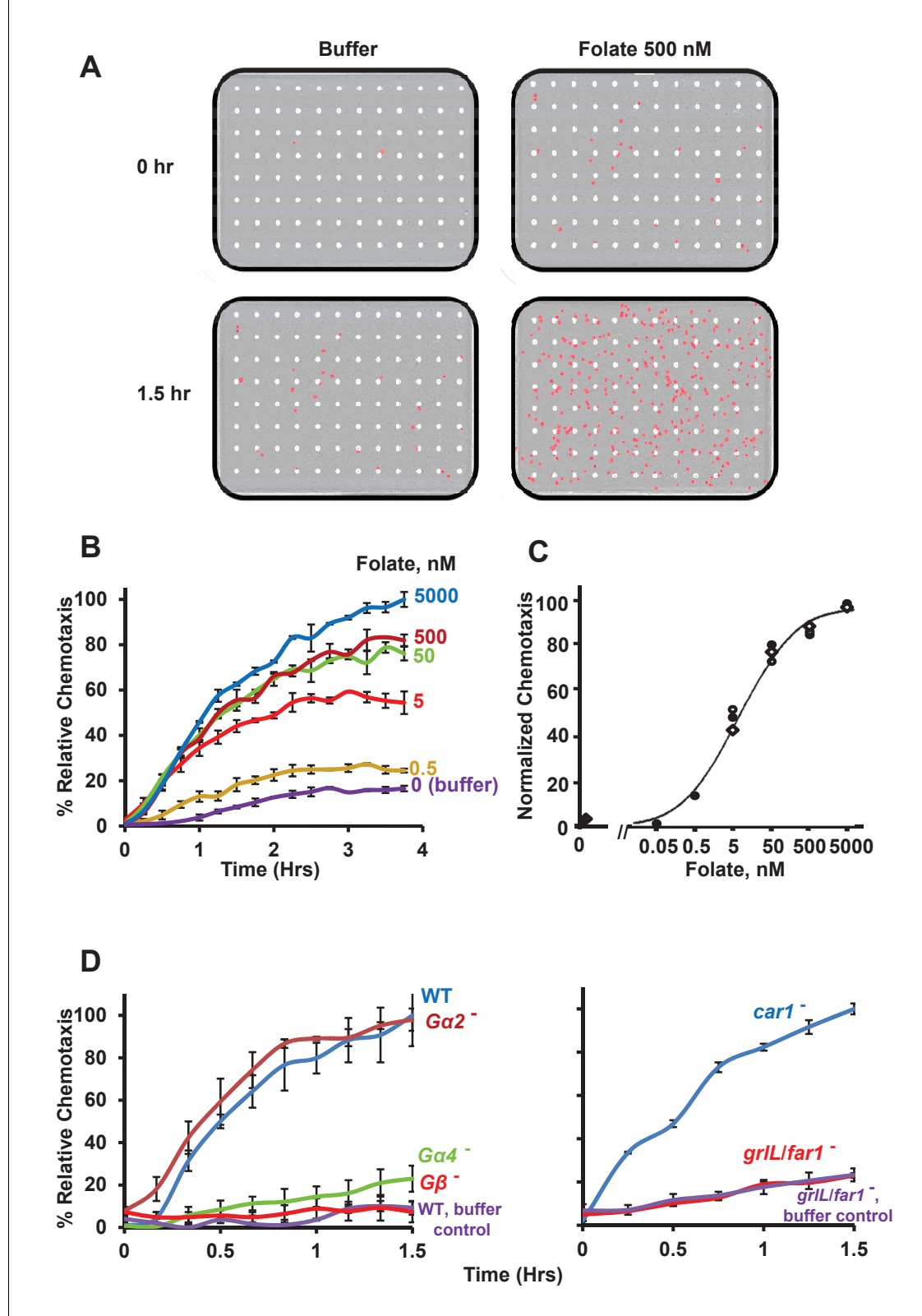

**Figure 1.** Chemotactic dose-response to folate. (**A**) Images of *Dictyostelium* (in red) that have migrated through membrane pores (in white) in response to either control buffer or the chemoattractant folate, at 0 and 1.5 hr. (**B**) Time-course quantification of *Dictyostelium* migration to various dose concentrations of folate. Relative chemotaxis is normalized to 5000 nM folate at 4 hr. Standard deviations are shown based upon three replicates. (**C**) Quantified dose-response for *Dictyostelium* chemotaxis to folate. Curve fit is for an $EC_{50}$ = 5 nM. Symbols presented are means from three separate

*Figure 1 continued on next page*

*Figure 1 continued*

experiments at 2 hr. (D) Time-course quantification of WT and Gα2-, Gα4-, Gβ-, car1-, and grlL/far1-null *Dictyostelium* migration to 500 nM folate or to buffer controls. Relative chemotaxis is normalized to WT (or car1-null) *Dictyostelium* at 1.5 hr; WT shows 10–20% higher relative chemotaxis to folate at 1.5 hr than does car1-null. Standard deviations are shown based upon three replicates.

The following figure supplement is available for figure 1:

**Figure supplement 1.** TaxiScan images of chemotaxing *Dictyostelium*.

folate (*Figure 1D*). Gα4 is generally considered essential for most folate responses, although limited, compensatory actions by other Gα proteins have been suggested (*Natarajan et al., 2000*); chemotaxis to folate is significantly reduced in cells lacking Gα4, but chemotaxis may not be eliminated (*Figure 1D*). The high sensitivity of this chemotaxis assay allows the detection of weak responses and additional pathway components. In contrast, cells lacking Gα2, a cAMP receptor-coupled G protein subunit (*Kumagai et al., 1989*), migrate to folate as robustly as do WT.

We also examined chemotaxis to folate in cells lacking or expressing specific chemoattractant GPCRs. The orphan receptor GrlL (*Hereld, 2005*; *Basu et al., 2013*) was recently defined as a GPCR for folate [fAR1 (*Pan et al., 2016*)], a conclusion that we have separately confirmed by comprehensive analyses of GPCRs expressed in growing *Dictyostelium* (NPM and ARK, in preparation). Cells lacking GrlL/fAR1 are unable to chemotax specifically to folate, in comparison to buffer controls [(*Pan et al., 2016*, *Figure 1D*)], whereas, car1-null cells that lack CAR1, the primary receptor for secreted cAMP and developmental chemotaxis (*McMains et al., 2008*; *Artemenko et al., 2014*; *Nichols et al., 2015*), respond robustly to folate signals. Thus, our data indicate that the migration assay to folate is both highly sensitive and highly specific.

## *Dictyostelium* chemotax to live gram positive and gram negative bacteria

Although the action of professional phagocytes to recognize and attack infecting bacteria is central to the innate immune response, quantifiable chemotaxis to live bacteria has been problematic. Growth-phase *Dictyostelium* migrate toward bacteria and engulf them for nutrient capture and the ability of *Dictyostelium* to detect chemoattractants at <0.5 nM suggested that the assay (see *Figure 1*)might be sufficiently sensitive to analyze directed migration to live bacteria.

*Bacillus subtilis,* gram positive bacteria, were washed extensively from media into buffer and loaded into the bottom well chamber at ~$10^4$–$10^8$ cells per compartment. *Dictyostelium* exhibited a precise dose-response to bacterial numbers (*Figure 2A*), as parallel to folate sensing (see *Figure 1B and C*). However, even at the limit of the system (~$5 \times 10^7$ bacteria/well), chemotaxis was not saturated; calculations suggest that >$5 \times 10^9$ bacteria/well would be required (see below).

*Dictyostelium* lacking the essential Gβ subunit for GPCR function were completely insensitive to the *Bacillus* (*Figure 2B*). Further, we observed only minimal chemotaxis to *Bacillus* lacking the folate-responsive Gα4 subunit (*Figure 2B*), whereas *Gα2*-null *Dictyostelium* were fully responsive (*Figure 2B*), suggesting that folate is the primary *Bacillus subtilis* chemoattractant (see below, Figures 5A and 7C) sensed by *Dictyostelium*.

We then further tested the ability of *Dictyostelium* to chemotaxis to gram negative bacteria, *Klebsiella planticola*, by similar chemotaxis assay. Interestingly the dose response to *Klebsiella planticola* was in a similar range to that of *Bacillus subtilis* (*Figure 2C*). Surprisingly however, *Dictyostelium* that lack either the folate-responsive Gα4 subunit or the cAMP-responsive Gα2

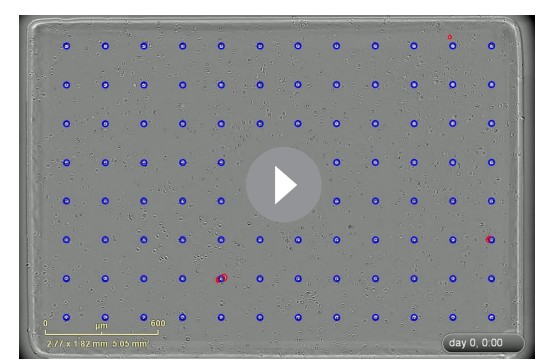

**Video 1.** Full 1.5 hr time-course of *Dictyostelium* migration through membrane pores toward a buffer control (see *Figure 1A*).

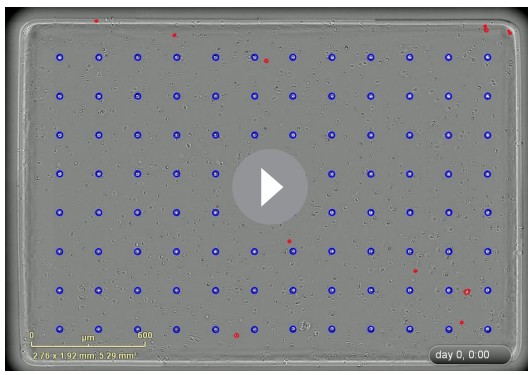

**Video 2.** Full 1.5 hr time-course of *Dictyostelium* migration through membrane pores toward 500 nM folate (see *Figure 1A*).

remained partially sensitive to *Klebsiella* (*Figure 2D*), suggesting that both folate and cAMP contribute to directional sensing. Although growing *Dictyostelium* express low levels of the chemoattractant receptor for cAMP, chemotactic migration of growth-phase *Dictyostelium* to cAMP has not been previously observed (*Veltman et al., 2014*). We, thus evaluated the ability of these growing *Dictyostelium* to chemotax directly to cAMP.

## *Dictyostelium* in logarithmic growth chemotax to cAMP

*Dictyostelium* secretes cAMP during development and responds chemotactically to promote cell aggregation at centers of cAMP signaling (*McMains et al., 2008*; *Artemenko et al., 2014*; *Nichols et al., 2015*). We had previously described conditions for the biochemical response of growth-phase *Dictyostelium* to cAMP (*Liao et al., 2013*), but chemotactic migration of growing *Dictyostelium* to cAMP has been an open question (*Veltman et al., 2014*). Interestingly, we detect rapid, robust, and highly sensitive migration of growth-phase WT *Dictyostelium* to cAMP (*Figure 3A,B*; $EC_{50}$ ~6 nM). This identical cell population is similarly chemotactic to folate (see below), indicating that they had not undergone a developmental shift from folate to cAMP response. Cells also remained chemotactic to cAMP when plated in defined growth media (data not shown), conditions which do not support a shift to a developmental pathway response (*Franke and Kessin, 1977*; *Basu et al., 2013*) or an upregulation in CAR1 expression (P. Jaiswal and ARK, in preparation). Cells lacking Gβ or Gα2 do not chemotax to cAMP, whereas cells lacking the folate-specific Gα4 are extremely responsive to cAMP (*Figure 3C*). In a reciprocal analysis of CAR1 and GrlL/fAR1 for cAMP detection, cells lacking GrlL/fAR1 show robust chemotactic response, whereas chemotaxis of *car1*-null cells to cAMP is extremely suppressed (*Figure 3C*).

## Chemotaxic responses to folate and cAMP participate in a common downstream activating network, but are not cross-adaptive

To define conditions that separate folate and cAMP responses, we examined chemotaxis by additive and competitive experiments. First, we compared chemotaxis toward either folate or cAMP, or toward combinations of both cAMP and folate, at varying concentrations. At sub-optimal (<50 nM) concentrations, folate and cAMP exhibit enhanced effects compared to either chemoattractant alone (*Figure 4A* and *Figure 4—figure supplement 1*). However, these effects may not be strictly additive. When a near-maximal dose (*i.e.* ~500 nM) of either folate or cAMP (see *Figures 1C* and *3B*) is applied, only minimal additional chemotaxis is observed when a second chemoattractant is included (*Figure 4A* and *Figure 4—figure supplement 1*). Thus, the folate and cAMP pathways act, at least partially, through common rate-limiting, downstream components for chemotaxis (see *Liao et al., 2013*).

When cells are exposed to saturating doses of a chemoattractant, their response pathways adapt and cells become insensitive for directed migration or biochemical activation, regardless of the gradient profile (*McMains et al., 2008*; *Artemenko et al., 2014*; *Nichols et al., 2015*). Thus, when cells (in the upper chamber) are plated in the presence of a saturating (5 µM) folate dose, they are inhibited from migrating toward non-saturating (50 nM) folate in the bottom well (*Figure 4B* and *Figure 4—figure supplement 2*). However, these folate-adapted cells, in the upper chamber, remain chemotactically responsive to low doses of cAMP in the bottom well. When the folate and cAMP treatments are reversed, we observed a reciprocal effect. Cells plated in the presence of saturating cAMP, in the upper chamber, do not migrate toward cAMP in the bottom well, but continue to chemotax to folate (*Figure 4B* and *Figure 4—figure supplement 2*), with perhaps slightly enhanced sensitivity. These results support our previous conclusions, based on biochemical experiments

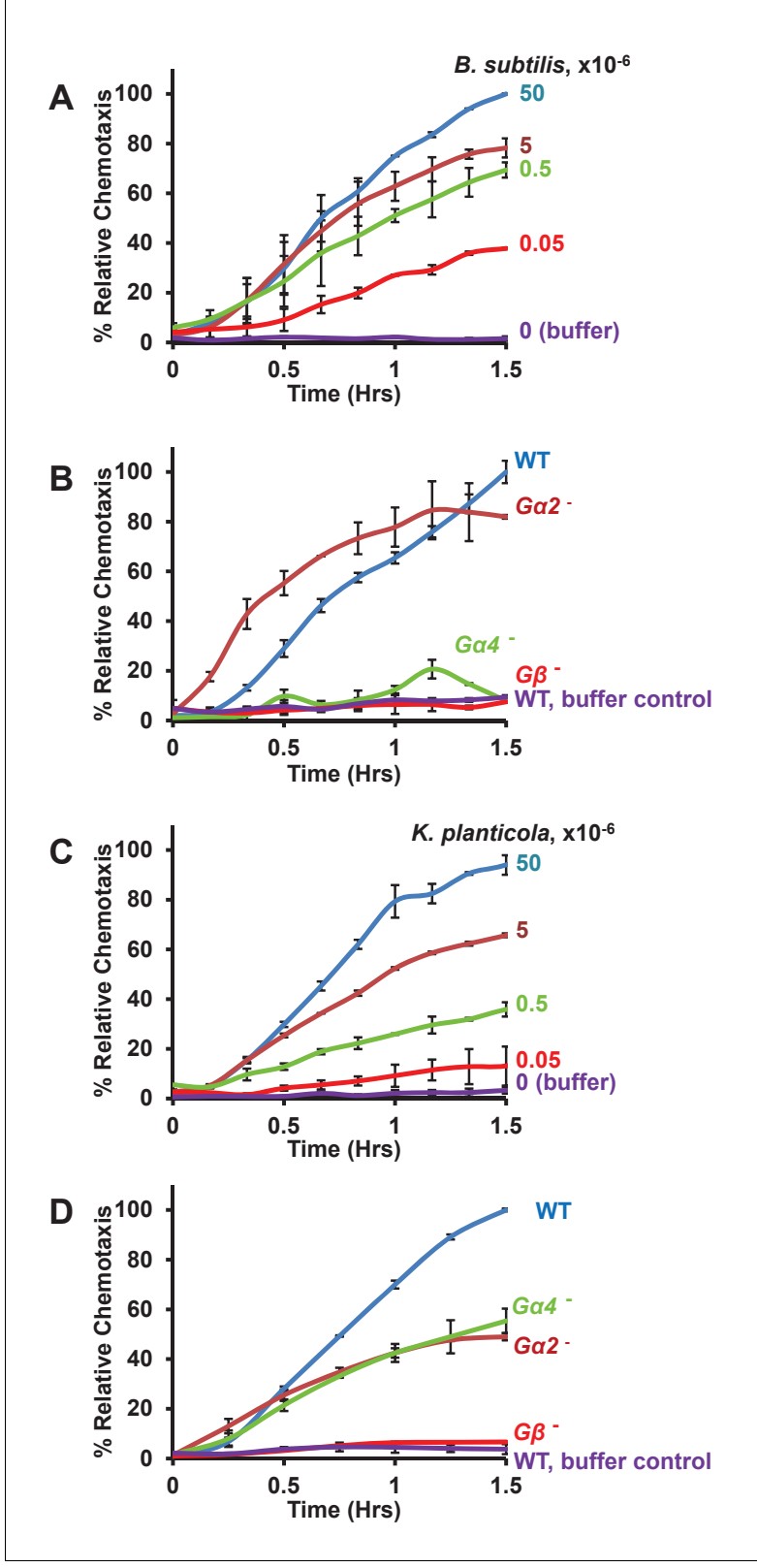

**Figure 2.** Chemotaxis to live bacteria. (**A**) Time-course quantification of WT *Dictyostelium* migration to various numbers of *Bacillus subtilis*. Relative chemotaxis is normalized to $5 \times 10^7$ bacteria at 1.5 hr. Standard deviations are shown based upon three replicates. (**B**) Time-course quantification of WT and *Gα2-*, *Gα4-*, and *Gβ*-null *Dictyostelium* migration to $5 \times 10^7$ *Bacillus subtilis* or to buffer controls. Relative chemotaxis is normalized to WT

*Figure 2 continued on next page*

*Figure 2 continued*

*Dictyostelium* at 1.5 hr. Standard deviations are shown based upon three replicates. (**C**) Time-course quantification of WT *Dictyostelium* migration to various numbers of *Klebsiella planticola*. Relative chemotaxis is normalized to 5 $\times$ $10^7$ bacteria at 1.5 hr. Standard deviations are shown based upon three replicates. (**D**) Time-course quantification of WT and *G$\alpha$2-*, *G$\alpha$4-*, and *G$\beta$*-null *Dictyostelium* migration to 5 $\times$ $10^7$ *Klebsiella planticola* or to buffer controls. Relative chemotaxis is normalized to WT *Dictyostelium* at 1.5 hr Standard deviations are shown based upon three replicates.

(*Liao et al., 2013*), that adaptation pathways for folate and cAMP function independently of one another and upstream of common downstream effector systems. As a further control, we show that the addition of folate or cAMP to cells in the upper chamber does not elicit a significant increase in basal motility, *via* ligand-activated chemokinesis (*Varnum and Soll, 1984*; *Kriebel et al., 2003*), to alter random movement toward a buffer control in the bottom well (*Figure 4B* and *Figure 4—figure supplement 2*). When both cAMP and folate are included at high concentrations in the upper well, we do observe a slight additional increase (to 20%) in these basal values. Certainly with time, chemo-attractants will diffuse into the lower chamber from the upper chamber, altering the gradient parameters. These data indicate that, although downstream components for folate and cAMP chemotaxis are shared, their adaptive mechanisms function upstream, *via* non-interacting pathways.

The competitive assay allowed us to further examine the relationship of *Bacillus* and *Klebsiella* to folate and cAMP. We observed only minimal chemotaxis to *Bacillus* of WT *Dictyostelium* using folate as a competitor, whereas WT cells plated in the presence of competing cAMP were fully responsive (*Figure 5A*). In contrast, competition by either folate or cAMP in the upper well only partially inhibited chemotaxis of WT *Dictyostelium* to *Klebsiella* (*Figure 5B*), confirming that both folate and cAMP contribute to directional sensing.

## Biochemical responses of *Dictyostelium* to bacterially secreted chemoattractants

To confirm the secretion of chemottractants from bacteria, we washed *Klebsiella planticola* from growth media into buffer at 22°C at 5 $\times$ $10^7$ bacteria/ml; supernatants were collected over time, and assayed against *Dictyostelium* for pERK2 activation, a kinase pathway that is activated downstream of both folate and cAMP receptors (*Meena and Kimmel, 2016*). No pERK2 activation was observed at zero-time, confirming that chemoattractants were not carried over from bacterial growth media (*Figure 6A*). However, a time-dependent increase in chemoattractant-simulated pERK2 was observed through 120 min of bacterial incubation (*Figure 6A*). Using an imunoassay to quantify cAMP, we show the secretion and accumulation of ~20–30 nM cAMP by 5 $\times$ $10^7$ *Klebsiella* cells in 2 hr. Due to the complex and heterogeneous structures of the folate/pterin class of compounds, it has not been possible to similarly confirm folate production levels. In addition, we were unable to detect cAMP secretion by *Bacillus* in a direct immunoassay, supporting our previous conclusions that cAMP was not a significant chemoattractant for *Bacillus* sensing.

To further correlate chemoattractant secretion to cellular response, we also assayed pERK2 activation using supernatants from different numbers of *Klebsiella planticola*. We observed the expected density-dependent level of secreted chemoattractants (*Figure 6B*). Interesting, although 5 $\times$ $10^6$ *Klebsiella planticola* will elicit a robust chemotactic response (see *Figure 2C*), they were unable to support pERK2 activation (*Figure 6B*), suggesting that chemotactic responses are more sensitive to activation than are individual biochemical pathways.

## Chemotaxis responses are more sensitive than are biochemical pathways

To better understand the quantitative chemotactic dose response of cells to folate and cAMP, we compared biochemical responses of growth-phase *Dictyostelium*, by measuring chemoattractant dose responses for relative phosphorylations of ERK2 and PKBR1, another folate/cAMP-activated kinase pathway (*Meena and Kimmel, 2016*). In standard activation assays, *Dictyostelium* are incubated at high cell densities (2–5 $\times$ $10^7$ cells/ml), leading to the rapid accumulation of very high levels of folate deaminase, which chemically inactivates folate (*Pan and Wurster, 1978*). At these cell

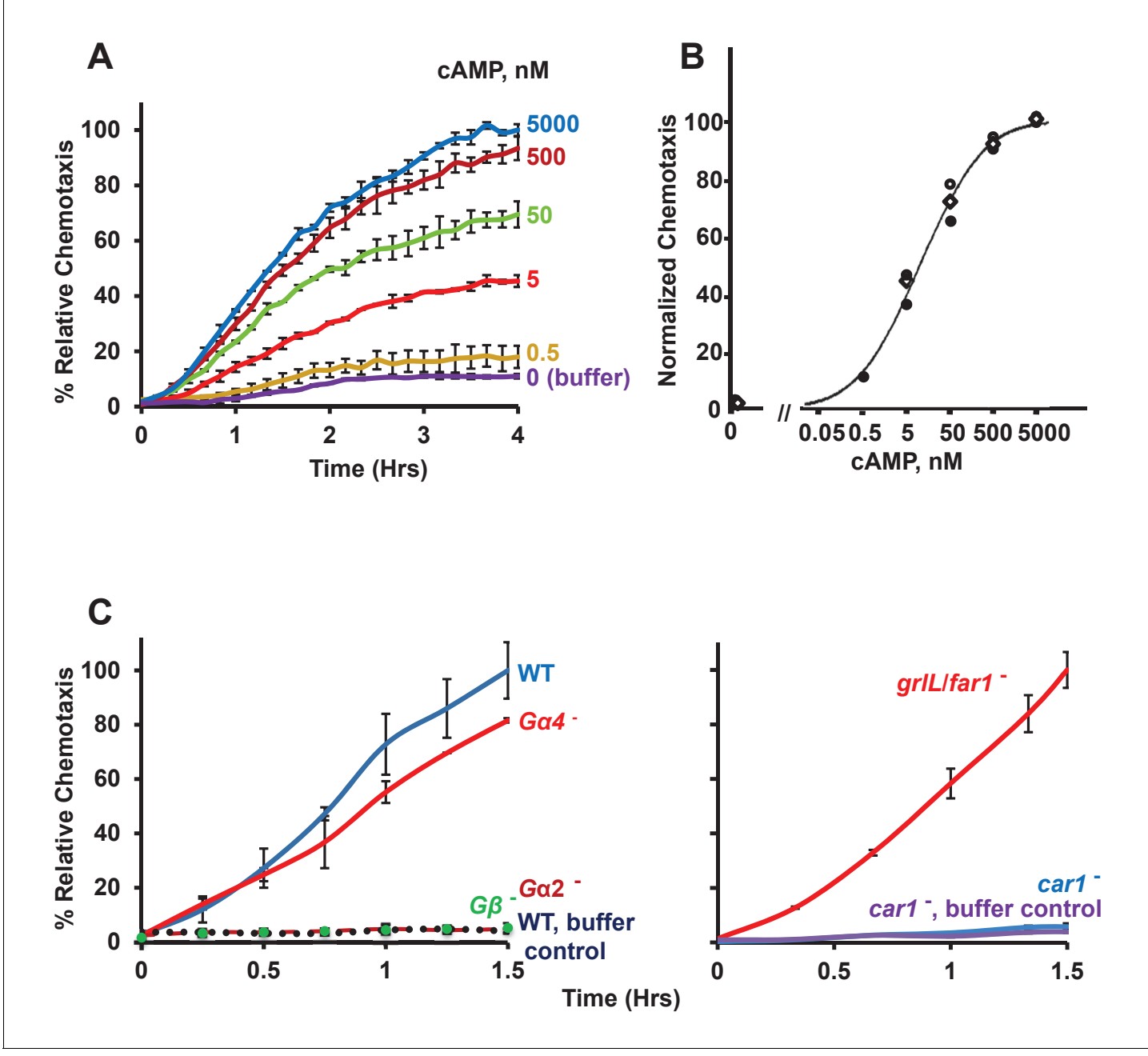

**Figure 3.** Chemotactic dose-response to cAMP. (**A**) Time-course quantification of *Dictyostelium* migration to various dose concentrations of cAMP. Relative chemotaxis is normalized to 5000 nM cAMP at 4 hr. Standard deviations are shown based upon three replicates. (**B**) Quantified dose-response for *Dictyostelium* chemotaxis to cAMP. Curve fit is for an $EC_{50}$ = 6 nM. Symbols presented are means from three separate experiments at 2 hr. (**C**) Time-course quantification of WT and *Gα2-*, *Gα4-*, *Gβ-*, *car1-*, and *grlL/folR*-null *Dictyostelium* migration to 50 nM cAMP or to buffer controls. Relative chemotaxis is normalized to WT (or *grlL/folR*-null) *Dictyostelium* at 1.5 hr; *grlL/far1*-null shows 10–20% higher relative chemotaxis to folate at 1.5 hr than does WT. Standard deviations are shown based upon three replicates.

densities, >20 nM folate can be inactivated per sec (NPM and ARK, in preparation), making biochemical response estimates of folate at low concentration doses highly variable. This is especially problematic for assays of pERK2, which has a slightly delayed response (30 s) compared to pPKBR1 (15 s). PDE, which inactivates cAMP, also accumulates to similarly high activities (*Brzostowski and Kimmel, 2006*). To minimize folate and cAMP inactivation, we assayed responses in more dilute

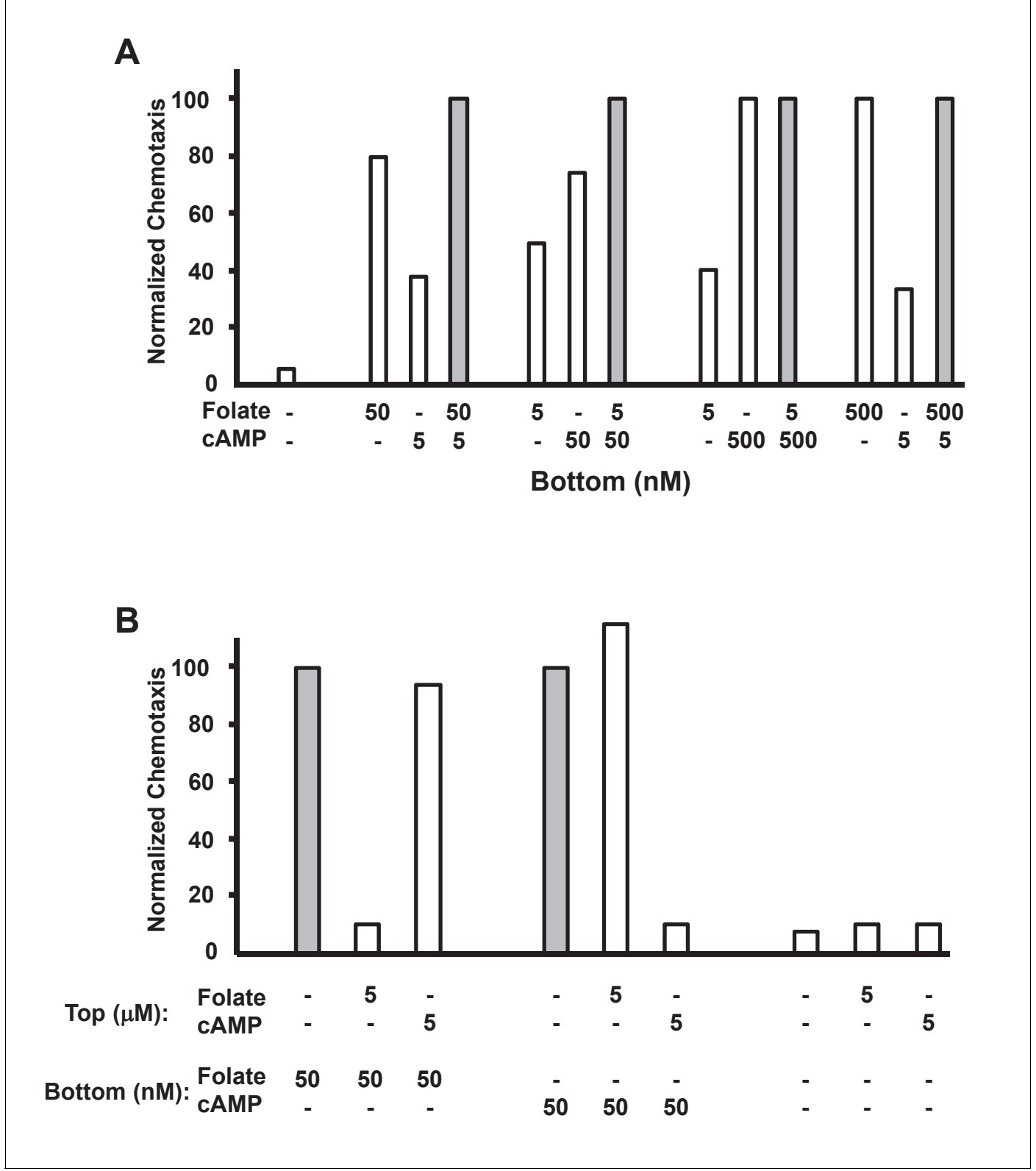

**Figure 4.** Additive and competitive effects of folate and cAMP to chemotaxis. (**A**) Quantified additive effects of folate and cAMP to chemotaxis. Folate and/or cAMP was used as a chemoattractant in the bottom chamber at varying concentrations (see *Figure 4—figure supplement 1*), as indicated. Normalizations were to maximum (shaded bar) chemotaxis within each 3-grouping. For SD statistics, etc. see *Figure 4—figure supplement 1*. (**B**) Quantified competitive effects of folate and cAMP to chemotaxis. Folate or cAMP was used as a chemoattractant in the bottom chamber, with or

*Figure 4 continued on next page*

*Figure 4 continued*

without a saturating (5 µM) folate or cAMP competitor in the top chamber (see *Figure 4B—figure supplement 2*), as indicated. Normalizations were to non-competed (shaded bar) chemotaxis within each 3-grouping. For SD statistics, etc. see *Figure 4—figure supplement 2*.

The following figure supplements are available for figure 4:

**Figure supplement 1.** Additive effects of folate and cAMP to chemotaxis.

**Figure supplement 2.** Competitive effects of folate and cAMP to chemotaxis.

volumes ($<10^6$ cells/ml), using cells adhered to plate surfaces and washed immediately prior to stimulation, to remove accumulated deaminase and PDE. Relative folate and cAMP responses for pPKBR1 and pERK2 were monitored by immune blotting (*Figure 6C*, *Figure 6—figure supplements 1* and *2*). For folate, the $EC_{50}$ for pPKBR1 and pERK2 are identical, ~45 nM, which is lower than previously noted but very similar to that of cAMP ($EC_{50}$ ~20/30/30 nM for pPKBR1/pERK2), when assayed simultaneously (*Figure 6C*). Nonetheless, for both cAMP and folate, *Dictyostelium* are 5–10 times more sensitive for chemotactic migration than for these biochemical pathway stimulations, consistent with chemotactic and bacterial responses for bacterial sensing (*Figures 2A, C*, *6A and B*).

## Phagocytosis does not depend upon chemoattractant sensing

At least three cellular processes share an interactive pathway for actin re-organization: chemotaxis, macropinocytosis, and phagocytosis (*Maniak et al., 1995*; *Hoeller et al., 2013*; *Veltman et al., 2014*; *Veltman, 2015*; *Junemann et al., 2016*). Although PIP$_3$-signaling, via localized PI3K activation, may be associated with each (*Servant et al., 2000*; *Veltman et al., 2014*; *Veltman, 2015*; *Hoeller et al., 2016*) and required for macropinocytosis (*Posor et al., 2013*; *Veltman et al., 2014*; *Veltman, 2015*), the pathway seems largely dispensable for both chemotaxis and phagocytosis of bacteria (*Hoeller and Kay, 2007*; *Hoeller et al., 2013*; *Schlam et al., 2015*). It is possible that folate-dependent activation of PI3K may sensitize membranes to facilitate phagocytic cup formation and that folate response may therefore, potentiate bacterial engulfment (*Pan et al., 2016*). However, bacterial-dependent growth rates, in shaking culture, of *Dictyostelium* lacking the folate receptor are only reduced by 20% over the long-term compared to WT cells (*Pan et al., 2016*), making a functional connection between folate receptor activation and phagocytosis unclear. We, thus, have carefully re-examined the interrelationship and dependency of phagocytic responses to chemoattractant GPCRs.

We used the IncuCyte system to image, in real-time, the engulfment of pHrodo-labelled *E. coli* (*Kapellos et al., 2016*) by *Dictyostelium*. Under these conditions, fluorescence is only detected intracellulary, within phagosome-derived acidified vacuoles (see *Figure 7A*). WT cells display rapid efficient phagocytosis, as measured by accumulation of total fluorescence through time (*Figure 7A*). Although cells lacking G$\beta$, a factor that had been previously defined as essential for phagocytosis in *Dictyostelium* (*Peracino et al., 1998*), exhibit only a minimal time-dependent increase in fluorescence, we did not observe a significant inhibition for phagocytosis in cells lacking G$\alpha$4, potentially suggesting that folate sensing is not a primary factor for phagocytosis. We then similarly compared rates of phagocytosis between cells that are deficient in either folate or cAMP chemoattractant GPCRs (see *Figures 1D* and *3C*) to equivalent strains that were engineered to re-express their corresponding GPCR, which have rescued chemoattractant responses (see *Figure 7B*).

As previously shown, *grlL/far1*-null cells do not chemotax to folate (*Figures 1D* and *7B*) and *car1*-nulls do not chemotax to cAMP (*Figures 3C* and *7B*). However, normal chemotactic responses could be restored upon re-expression of WT GrlL/fAR1 or CAR1 in their corresponding deficient strains (*Figure 7B*). These genetically matched cell lines, thus, present ideal models to assess the role of chemoattractant receptor sensing in phagocytosis. We do not observe significant differences in the rates of phagocytosis of pHrodo-labelled *E. coli* between *grlL/far1*-null and the GrlL/fAR1-expressing rescued cells (*Figure 7B*); likewise, *car1*-nulls and CAR1-expressing cells exhibit identical phagocytic properties (*Figure 7B*). These data indicate that phagocytosis per se is not substantially regulated

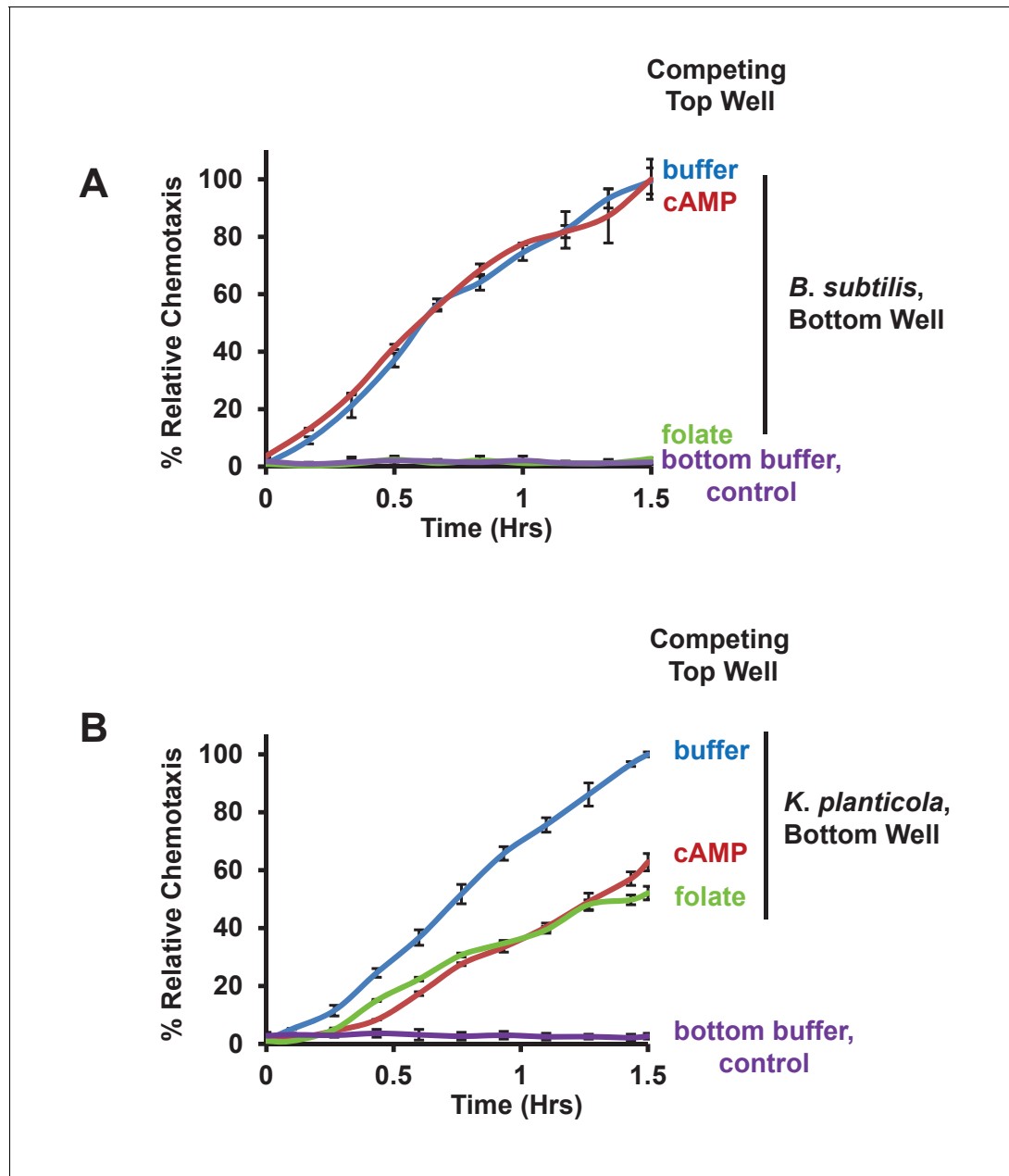

**Figure 5.** Chemotaxis to live bacteria in the presence of competing chemoattractants. (A) Time-course quantification of WT *Dictyostelium* migration to $5 \times 10^7$ *Bacillus subtilis* or to buffer controls in the presence of competing levels of 5 μM folate or 5 μM cAMP in the top well, as indicated. Relative chemotaxis is normalized to WT *Dictyostelium* at 1.5 hr without any competitor. Standard deviations are shown based upon three replicates. (B) Time-course quantification of WT *Dictyostelium* migration to $5 \times 10^7$ *Klebsiella planticola* or to buffer controls in the presence of competing levels of 5 μM folate or 5 μM cAMP in the top well, as indicated. Relative chemotaxis is normalized to WT *Dictyostelium* at 1.5 hr without any competitor. Standard deviations are shown based upon three replicates.

by chemoattractant receptors nor is it markedly influenced by their dependent downstream activations.

It may be argued that pHrodo-labelled *E. coli* may be compromised for viability and chemoattractant production, and, therefore, not ideal to fully assess receptor-sensitive pathways. Further, potentially the secretion of multiple chemoattractants (*e.g.* folate and cAMP) by bacteria may complicate data interpretations for functional responses of *grlL/far1*- or *car1*-null cells. We, thus, modified the bacterial labeling conditions to preserve viability and additionally used *Bacillus subtilis*, where the

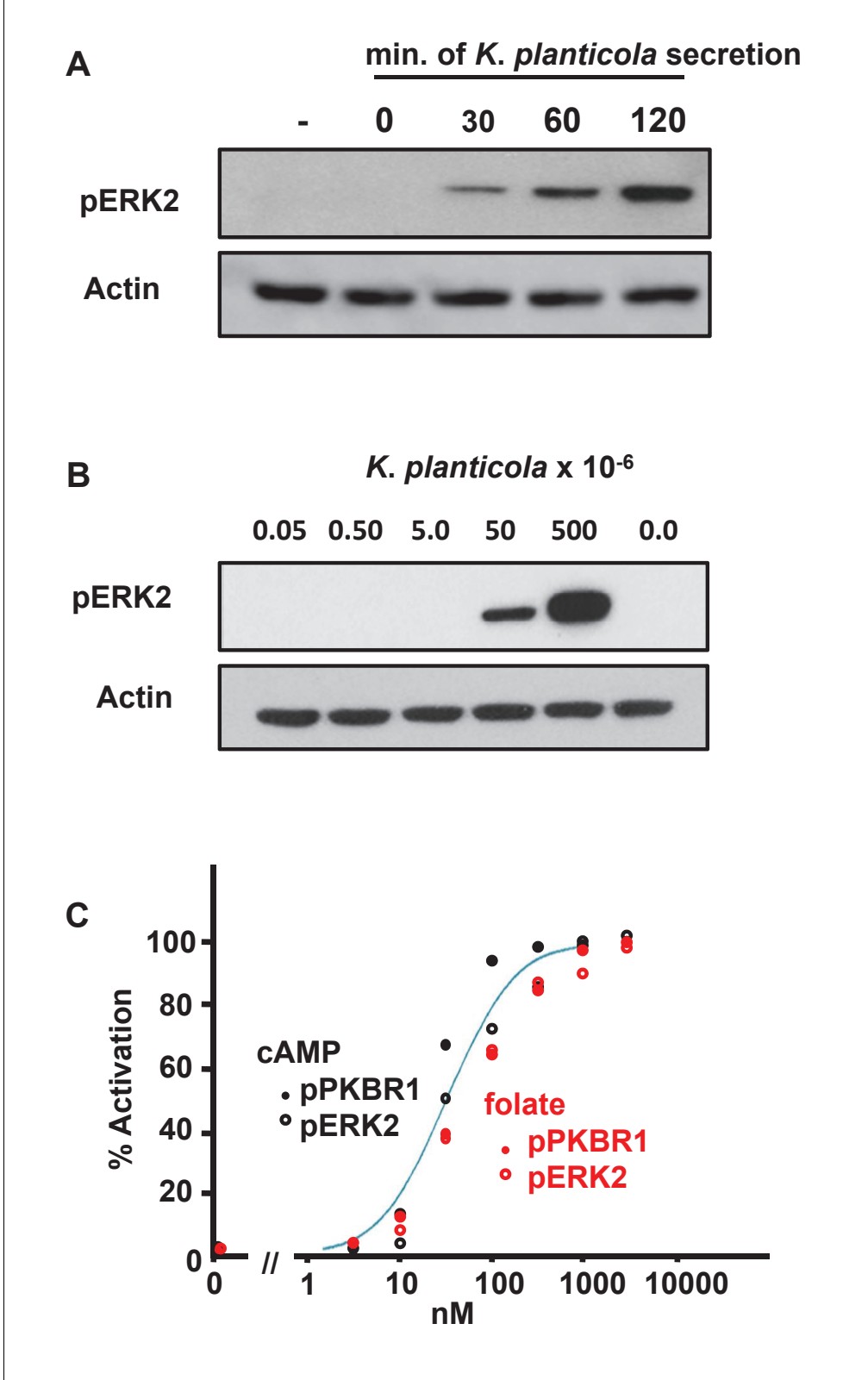

**Figure 6.** Biochemical responses of *Dictyostelium* to bacterially secreted chemoattractants. (**A**) Time-course accumulation of secreted chemoattractants by *Klebsiella planticola*, at $5 \times 10^7$ bacteria/ml, as assayed by pERK2 activation. *Dictyostelium* were stimulated with supernatants from *Klebsiella planticola* cultures incubated for the times indicated. Proteins were immunoblotted to α-pERK2 to confirm appropriate time courses for activation and adaptation. 30 s samples were separately immunoblotted for pERK2 activation and normalizing actin. (**B**) Density-dependent effect of secreted

*Figure 6 continued on next page*

*Figure 6 continued*

chemoattractants by *Klebsiella planticola*, as assayed by pERK2 activation. *Klebsiella planticola* were washed from media and incubated at $5 \times 10^9$ cells/ml in buffer for 2 hr. The bacterial supernatant was collected, and the volumes equivalent to the cell numbers indicated were used to stimulate *Dictyostelium*. Proteins were immunoblotted to α-pERK2 to confirm appropriate time courses for activation and adaptation. 30 s samples were separately immunoblotted for pERK2 activation and normalizing actin. (C) Quantified dose-response for the phosphorylations of PKBR1, at 15 s, and ERK2, at 30 s, to folate (see *Figure 6—figure supplement 1*) and cAMP (see *Figure 6—figure supplement 2*), and normalized to actin. Curve fit is for an $EC_{50}$ = 30 nM.

The following figure supplements are available for figure 6:

**Figure supplement 1.** Biochemical dose-responses to folate.

**Figure supplement 2.** Biochemical dose-responses to cAMP.

primary chemoattractant is folate and additional responses due to co-secretion of cAMP do not occur (see *Figures 2B* and *5A*). The labeled *Bacillus subtilis* population was then used in assays for both chemotaxis and phagocytosis of *grlL/far1*-null and GrlL/fAR1-expressing cells (*Figure 7C*). GrlL/fAR1-expressing cells chemotax to labeled *B. subtilis*, whereas the *grlL/far1*-nulls were completely insensitive. Yet, both cell lines exhibit identical parameters for phagocytosis, thus separating an essential GrlL/fAR1 dependency for chemotaxis to folate-secreting cells, from a functional role in phagocytosis.

Although the folate GPCR is essential for chemotaxis to folate-secreting bacteria, this receptor does not appear to have a significant function in the phagocytosis of these same bacteria. However, it is possible that chemoattractant stimulation may potentiate phagocytic processes. To test this experimentally, we stimulated WT *Dictyostelium* for 1 hr with either 50 nM cAMP or 50 nM folate pulses at 6 min intervals, which synchronizes cells for cyclic activation/adaptation and mimics chemoattractant wave propagation and response in vivo (*Meena and Kimmel, 2016*); phagocytosis of pHrodo-labelled *E. coli* by these cells were then compared to non-stimulated WT controls. No differences in phagocytic rates were observed, regardless of treatment (*Figure 8*), indicating that chemotactic-activation does not inherently potentiate cells for phagocytic functions.

## Folate ligands on particle surfaces do not stimulate or inhibit phagocytic processes

Molecules on particle surfaces are well known recognition marks for phagocytes, although it has not been clear if surface-immobilized chemoattractant ligands are recognized by cognate receptors to facilitate phagocytosis. We, thus, compared the ability of WT *Dictyostelium* and cells lacking the folate receptor to phagocytose 1 μm latex beads with and without folate surface ligands.

It is well established that *Dictyostelium* can rapidly engulf synthetic latex beads (see *Vogel et al., 1980*). Here, we first labelled 1 μm latex beads with the pH sensitive pHrodo dye (see *Figure 9—figure supplement 1*) and established conditions for quantitative bead phagocytosis, based upon an increase in rhodamine fluorescence as they are incorporated within phagosome-derived acidified vesicles. WT and *grlL/far1*-null cells show identical rates of phagocytosis to pHrodo-beads (*Figure 9A*).

We then secondarily labelled the surfaces of pHrodo-beads with Folate-FITC, anchoring the FITC moiety to free amines on the surface of the pHrodo-beads. We estimate that each bead displays, on average, ~20,000 folate surface ligands and there is a near 100% coordinate labeling of pHrodo and Folate-FITC fluorescence markers at the bead surfaces (see *Figure 9—figure supplements 1* and *2*). Identical rates of phagocytosis were observed with WT cells using either pHrodo- or Folate-FITC/pHrodo-beads as a substrate (*Figure 9B*), indicating that WT cells, which express folate receptors, do not differentially recognize folate-coated from non-coated beads for functional phagocytosis. Finally, we demonstrate that WT cells and cells lacking the chemotaxis folate receptor phagocytose Folate-FITC/pHrodo-beads with identical facility (*Figure 9C*). These data further indicate that chemo-sensing of ligands on particle surfaces does not significantly contribute to phagocytic processes.

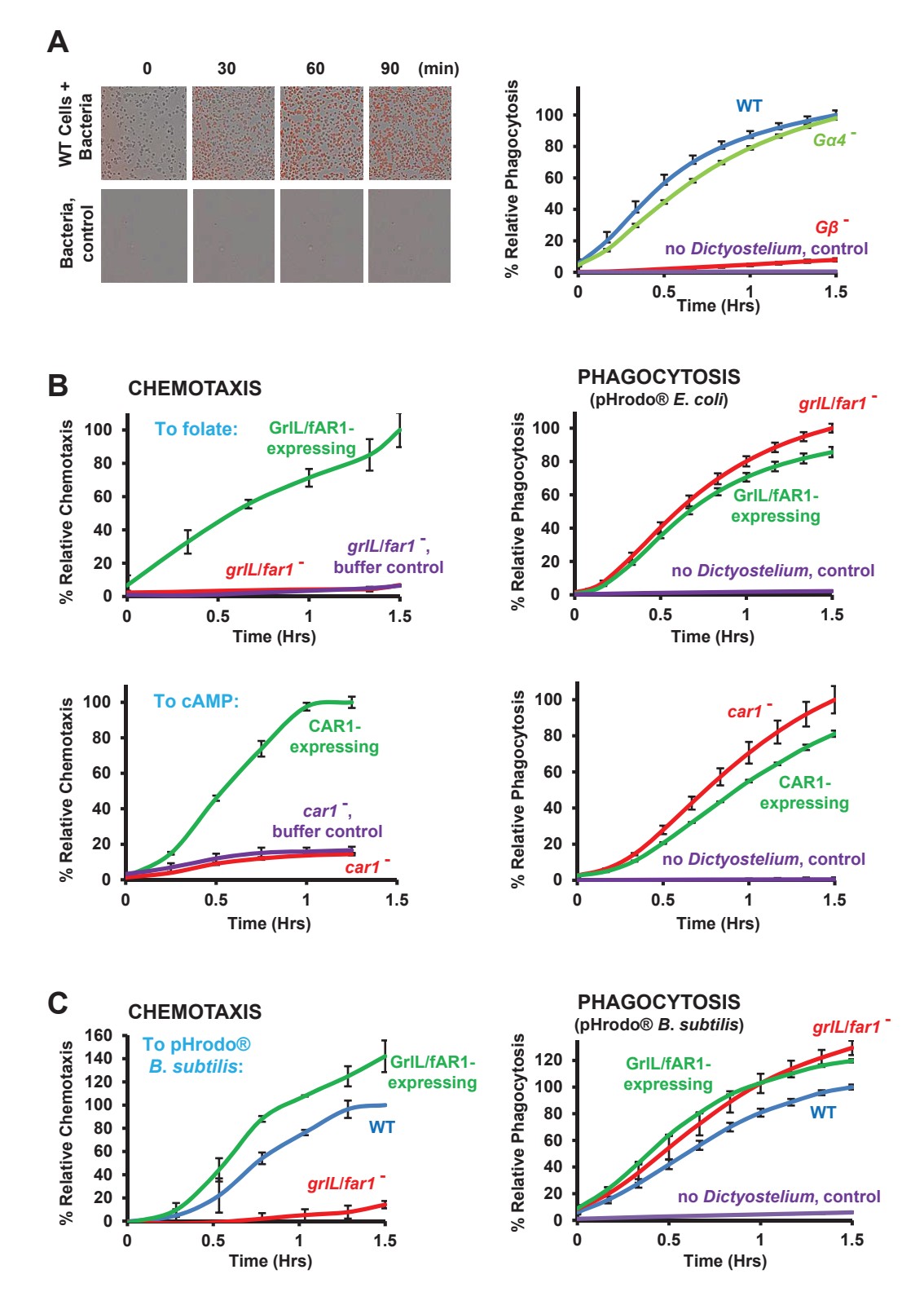

**Figure 7.** Loss of GPCRs in *Dictyostelium* impairs chemotaxis but not phagocytosis. (**A**) Fluorescent images of pHrodo-labelled *E. coli* (in red) engulfed by WT *Dictyostelium* over time. Time-course quantification of WT and *Gα4-* and *Gβ*-null fluorescence normalized to WT at 1.5 hr. Standard deviations are shown based upon three replicates. (**B**) FOR CHEMOTAXIS: Time-course chemotaxis quantification of *grlL/far1*- and *car1*-null cells and their respective GPCR-expressing cell lines to 500 nM folate or 500 nM cAMP (as indicated), with buffer controls. Relative chemotaxis is normalized to the

*Figure 7 continued on next page*

*Figure 7 continued*

respective GPCR-expressing cell line at 1.5 hr. Standard deviations are shown based upon three replicates. FOR PHAGOCYTOSIS: Time-course phagocytosis quantification of pHrodo-labelled *E. coli* by *grlL/far1*- and *car1*-null cells and their respective GPCR-expressing cell lines, normalized to the respective null cells at 1.5 hr. Standard deviations are shown based upon three replicates. (**C**) FOR CHEMOTAXIS: Time-course chemotaxis quantification of WT, *grlL/far1*-null and GrlL/fAR1-expressing cells to live pHrodo-labelled *B. subtilis*. Relative chemotaxis is normalized to WT at 1.5 hr. Standard deviations are shown based upon three replicates. FOR PHAGOCYTOSIS: Time-course phagocytosis quantification of live pHrodo-labelled *B. subtilis* by WT, *grlL/far1*-null, and GrlL/fAR1-expressing cells, normalized to WT at 1.5 hr. Standard deviations are shown based upon three replicates.

## Discussion

Chemotactic response of developing *Dictyostelium* to cAMP has been a paradigm for understanding macrophage and neutrophil migratory behavior for immune function (*Jin et al., 2008*; *McMains et al., 2008*; *Artemenko et al., 2014*; *Nichols et al., 2015*). Growing *Dictyostelium* are also highly chemotactic and, as active phagocytes during this part of their life cycle, present an excellent alternative to extend and unify analyses for chemotaxis to and phagocytosis of bacteria.

We have established conditions that allow the high sensitive detection/migration to bacteria by growth-phase *Dictyostelium* and identify both folate and cAMP as active bacterially secreted chemo-attractants, at <0.5 nM sensitivity. Although we had previously shown that *Dictyostelium* express

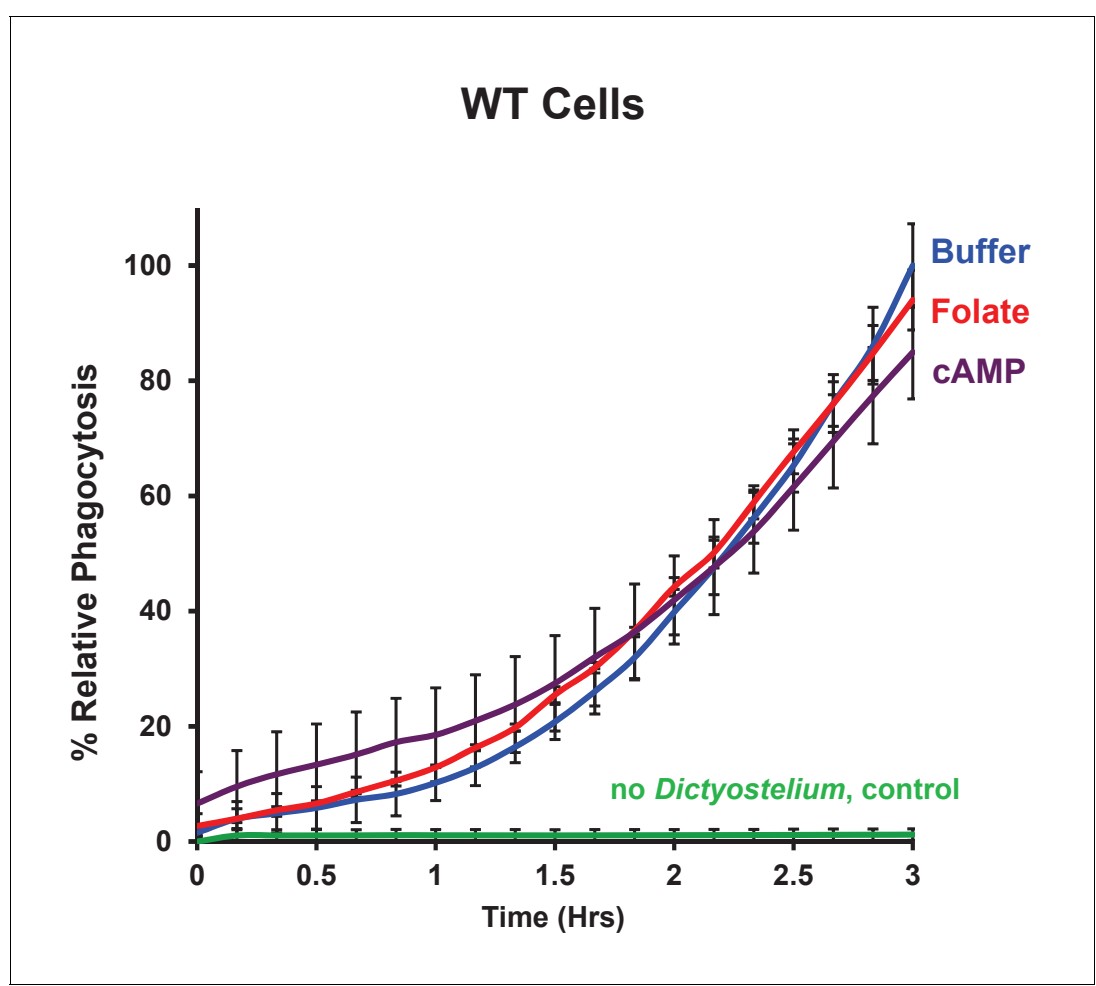

**Figure 8.** Chemoattractant-stimulation does not activate phagocytosis. Time-course phagocytosis quantification of pHrodo-labelled *E. coli* by control WT or chemotactically-stimulated WT cell lines, normalized to buffer treated control cells at 3 hr. Standard deviations are shown based upon three replicates.

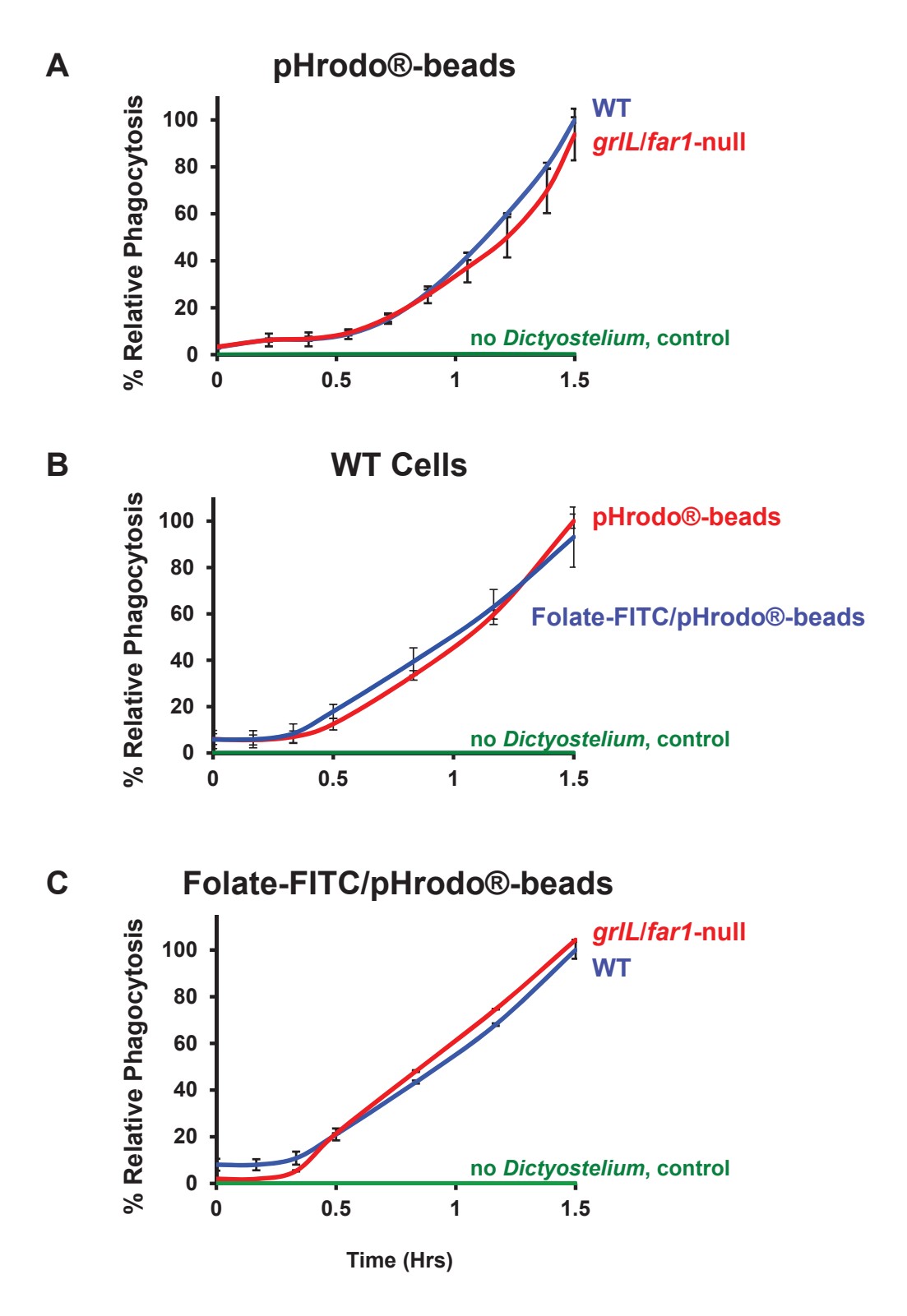

**Figure 9.** Folate surface coating of latex beads does not stimulate their phagocytosis by *Dictyostelium*. (**A**) Time-course phagocytosis quantification of pHrodo-labelled latex beads (see *Figure 9—figure supplement 1*) by WT or *grlL/far1*-null cells, normalized to WT cells at 1.5 hr. Standard deviations are shown based upon three replicates. (**B**) Time-course phagocytosis quantification of pHrodo-labelled and Folate- FITC/pHrodo-labelled latex beads (see *Figure 9—figure supplements 1* and *2*) by WT *Dictyostelium*, normalized to pHrodo-beads at 1.5 hr. Standard deviations are shown based upon

*Figure 9 continued*

three replicates. (C) Time-course phagocytosis quantification of Folate-FITC/pHrodo-labelled latex beads (see *Figure 9—figure supplement 2*) by WT or *grlL/far1*-null cells, normalized to WT cells at 1.5 hr. Standard deviations are shown based upon three replicates.

The following figure supplements are available for figure 9:

**Figure supplement 1.** Co-localization of pHrodo and Folate-FITC labels to all latex beads.

**Figure supplement 2.** Co-localization of pHrodo and Folate-FITC labels to all latex beads.

cAMP receptors during growth at sufficient levels for biochemical response to cAMP (*Liao et al., 2013*), migratory behavior of growing *Dictyostelium* to cAMP had been disputed (*Veltman et al., 2014*). Nonetheless when cAMP signal concentrations are applied at >100 fold levels, chemotactic response to cAMP during growth is similarly sensitive to that of folate, even using other assay systems (*Figure 1—figure supplement 1*). The ability of growing *Dictyostelium* to respond to bacterially-secreted cAMP may provide an additional signal for nutrient sensing, which would suppress the transitional switch from growth to development, where starved *Dictyostelium* utilize chemotaxis to secreted cAMP as a mode for multi-cell recognition and aggregate formation.

Several parameters that influence the shape of individual chemoattractant gradients are also critical to enhance migratory response. For folate and cAMP, the diffusion from a concentrated signal in the bottom well to a 'zero' signal in the upper chamber creates an initial highly steep gradient that would be expected to diminish with time, even as total chemoattractant levels in the upper chamber rise. The secretion and accumulation of chemoattractant de-activating enzymes (deaminase for folate; PDE for cAMP) by *Dictyostelium* serve to attenuate signal levels in close cellular proximity, thereby strengthening the concentration of the gradient and reinforcing directional migration (*Mackenzie et al., 2016*; *Tweedy et al., 2016*). Other systems may utilize analogous mechanisms to amplify signaling gradients through ligand degradation, sequestration, or de novo synthesis (*Muinonen-Martin et al., 2014*; *Scherber et al., 2012*; *Venkiteswaran et al., 2013*; *Donà et al., 2013*; *Boldajipour et al., 2008*). Also with time, as large sub-populations of *Dictyostelium* migrate up the gradient (*i.e.* toward the lower chamber), these secreted ligand-degrading enzymes would eliminate chemoattractant signals at the rear of the moving cells, shielding lagging cells from recruitment for activation and motility (*Mackenzie et al., 2016*; *Tweedy et al., 2016*). For migration to bacteria, the continued secretion and accumulation of chemoattractants in the bottom chamber would create a dynamically changing gradient shape, with an exponentially increasing stimulus of the phagocytes, as they approach the source for chemoattractant secretion.

At least 6 signaling arms are activated downstream of chemoattractant receptors in *Dictyostelium* (*Van Haastert and Veltman, 2007*; *McMains et al., 2008*; *Artemenko et al., 2014*). Although low level (*e.g.* 1 nM) chemoattractant stimulation may elicit too weak a response to be detectable in an individual biochemical read-out, the collective and coordinated activities of multiple signaling paths are sufficient to initiate directed migration, at <0.5 nM. Since chemotactic signaling networks are shared among *Dictyostelium*, macrophages, and neutrophils, it is likely that similar interplays function during immune response to ensure high sensitivity for chemotaxis. We suggest that bacterial detection may be further enhanced under conditions of even more limited (*e.g.* 0.1 nM) signal production, where phagocytes migrate toward multiple bacterially-secreted chemoattractants that can act cooperatively at low concentrations. Of note, bacterial chemoattractant production levels and gradients were assayed at conditions (*e.g.* 22°C, in buffer) optimal for *Dictyostelium*; it may be anticipated that <0.1x bacterial numbers could be detected within a mammalian physiological mileu (*e.g.* 37°C).

At the other end of the concentration spectrum (>100 µM), *Dictyostelium* chemotaxis is diminished (data not shown). At these concentrations, there is ligand-specific down regulation of GrlL/fAR1 (NPM and ARK, in preparation) and CAR1 protein and mRNA expression (*Kimmel, 1987*; *Klein et al., 1987*; *Louis et al., 1993*). We suggest that reduced chemotaxis of growing *Dictyostelium* toward areas of very high chemoattractant concentration may be an avoidance measure, since *Dictyostelium* have reduced viability in the presence of extreme bacterial density [(*Nasser et al., 2013*) NPM and ARK, in preparation].

The ability of individual growing *Dictyostelium* to simultaneously respond to multiple chemoattractants permitted analyses of both additive and competitive effects. Although migratory responses to multiple chemoattractants may be enhanced at low concentrations in comparison to a single chemoattractant, such cooperating effects are not observed at saturating doses, indicating that pathways activated downstream of the separate receptors must function in common. At globally applied, saturating levels of ligand, migratory cells lose directional response (*McMains et al., 2008*; *Artemenko et al., 2014*; *Nichols et al., 2015*). These cells undergo an initial rapid stimulatory activation, but then become unresponsive and globally adapt multiple receptor-mediated pathways (*McMains et al., 2008*; *Artemenko et al., 2014*; *Nichols et al., 2015*). Quiescence persists if extracellular ligand concentrations remain saturated. Identification of adaptive mechanisms has been elusive. Although certain mathematical models have suggested that adaptive pathways must act globally and downstream of GPCRs, we have demonstrated otherwise (*Liao et al., 2013*). Cells adapted to one chemoattractant remain fully responsive to a different stimulus, indicative of ligand selective adaptation (*Liao et al., 2013*). We have also shown that part of the adaptive network is mediated by GPCR phosphorylation, induced by ligand-specific interactions (*Brzostowski et al., 2013*). We further confirm, here, that cells plated in the presence or absence of saturating (adapting) folate (or reciprocally cAMP) are equally chemotactic to cAMP (or folate), indicating that adaptive mechanisms are not shared between the folate and cAMP pathways.

These data suggest a mechanistic model for chemotactic response under varying signaling parameters that maximize phagocytic behavior. Activations by low-dose stimulation with multiple chemoattracts will amplify gradient detection and low density bacterial sensing. However, cells that have become desensitized though ligand saturation by one chemoattractant (*Wu and Lin, 2011*; *Liao et al., 2013*), remain fully responsive to activation by other bacterially secreted chemoattractants.

For phagocytes, chemotaxis is a means to engage prey. Both chemotaxis and phagocytosis require polymerized actin at defined membrane surfaces, and there has been conjecture if such interconnected processes act antagonistically and/or cooperatively (*Maniak et al., 1995*; *Heinrich and Lee, 2011*; *Hoeller et al., 2013*; *Veltman et al., 2014*; *Veltman, 2015*; *Junemann et al., 2016*; *Jones et al., 2016*). Actin-associated $PIP_3$ levels can accumulate at the leading membrane edge of cells that are migrating within chemoattractant gradients (*Parent et al., 1998*; *Meili et al., 1999*; *Servant et al., 2000*). Similarly, $PIP_3$ membrane patches have been shown to co-localize with phagocytic cups (*Hoeller et al., 2013*; *Pan et al., 2016*). In this connection, it has been suggested that chemoattractant-mediated receptor activation of PI3K may coordinate localized accumulation of $PIP_3$ to promote actin polymerization and phagocytic cup formation (*Pan et al., 2016*). Workers have argued that cells lacking the folate chemoattractant receptor have diminished $PIP_3$-signaling and phagocytosis compared to WT, and connected a direct dependency for folate-mediated $PIP_3$ localization with phagocytic cup formation (*Pan et al., 2016*). These conclusions, however, are not fully compatible with our data or with that of other published work.

We systematically compared the chemotactic ability of cells lacking the folate receptor (*grlL/far1*-nulls) with *grlL/far1*-null cells that had been engineered to re-express GrlL/fAR1 and demonstrated the absence of folate chemotaxis by *grlL/far1*-nulls and the definitive rescue of chemotaxis to folate and to folate-secreting *Bacillus subtilis* by re-expression of GrlL/fAR1. Regardless of chemotactic ability, both cell lines exhibited identical parameters for bacterial phagocytosis. Similarly, although *car1*-null cells are chemotactically insensitive to cAMP, they are as active as chemotactically-positive, CAR1-expressing cells for bacterial phagocytosis. To avoid complications of folate- and cAMP-receptor co-dependence on the phagocytoisis of bacteria that secrete both chemoattractants, we separately confirmed the chemotactic and phagocytic properties of *grlL/far1*-null and GrlL/fAR1-expressing cells using *B. subtilis*, whose primary secreted chemoattractant is folate. We do not detect a significant contribution of chemo-sensing activation *via* GrlL/fAR1 or other chemoattractant receptors to mediate phagocytosis by adherent *Dictyostelium*. Neither did we detect differences in phagocytosis between chemotactically activated or quiescent WT *Dictyostelium*. Finally, we showed that WT *Dictyostelium* or cells that are deficient for the folate chemoattractant receptor phagocytose folate-coated and non-coated, control latex beads to equivalent rates.

Our conclusions are supported by additional evidence. We did not detect differences in bacterial phagocytosis when comparing $G\alpha4$-nulls to WT controls, consistent with other observations (*Peracino et al., 1998*). Although a partial dependency on $G\alpha4$ for phagocytosis of latex beads has

been suggested in other studies (*Gotthardt et al., 2006*), these data were derived independently of the presence of folate. It is well appreciated that G$\beta$ in *Dictyostelium* has a significant role for bacterial phagocytosis (*Peracino et al., 1998*; *Gotthardt et al., 2006*), and it is argued that G$\beta$ becomes activated downstream of the folate receptor for control of phagocytosis (*Pan et al., 2016*). However, where cells lacking G$\beta$ show a >3x reduced growth rate on bacteria in shaking culture (*Peracino et al., 1998*), the published growth rate of *far1*-null cells on bacteria is reduced by only ~0.2x compared to controls (*Pan et al., 2016*), suggesting a limited absolute role for folate receptor signaling in long-term bacterial phagocytosis. A universal requirement for chemoattractant receptor activation of G proteins in the regulation of phagocytosis is, also, not supported by other studies. Mammalian cells lacking all G$\beta$ subunits have similar rates of phagocytosis compared to controls (*Hwang et al., 2005*). We recognize that G$\beta$ subunits localize with phagosomes (*Desjardins et al., 1994*; *Gotthardt et al., 2006*), although it has also been suggested that G$\beta$ may have functions that are independent of GPCR activation (*Hoeller et al., 2016*).

We have not examined the role of GrlL/fAR1 in PIP$_3$ production or PIP$_3$ association with phagocytic cup formation, but their biological connections to folate response appear to be limited. *pi3k1-5*-null cells accumulate <10% of WT PIP$_3$ levels, but show no inhibition of phagocytosis of bacteria, growth on bacteria, or chemotaxis, compared to WT (*Hoeller and Kay, 2007*; *Hoeller et al., 2013*). Whereas PIP$_3$-signaling may be dispensable for phagocytosis of bacteria (see *Schlam et al., 2015*), these data do not exclude the possibility that folate stimulated PIP$_3$-patch formation may potentiate phagocytic cup formation to some extent. In WT, PIP$_3$-free pseudopods primarily orient towards folate during migration (*Veltman et al., 2014*; *Veltman, 2015*), suggesting that if PIP$_3$-labeled cellular protrusions participate in phagocytic cup formation, they are not initiated *via* folate signaling. Macropinocytosis, a process for fluid-phase uptake that requires PI3K/PIP$_3$-signaling (*Hoeller et al., 2013*; *Posor et al., 2013*), is functionally antagonistic to chemotaxis (*Veltman et al., 2014*; *Veltman, 2015*). Thus, PIP$_3$ production interferes with pseudopod orientation during chemotaxis of growing cells to folate, further disconnecting PI3K/PIP$_3$-signaling from chemotaxis and phagocytosis. Indeed, in other processes phagocytic cup and chemotaxis leading edge formation are competitive events (*Maniak et al., 1995*).

We carefully, first, defined widely sensitive chemotactic conditions to pure ligands and bacteria, and followed phagocytic assays of cells adhered to culture plates in real time, with near linear activity through >2 hr, and coupled these parameters to assess interrelationships between bacterial chemosensing and directed phagocytic behavior. Our data demonstrate efficient phagocytosis that is independent of chemo-sensing and chemotaxis. Other approaches using mammalian neutrophils have also defined incubation conditions that will suppress chemotaxis without affecting phagocytosis (*Heinrich and Lee, 2011*; *Mankovich et al., 2013*). Still, we recognize that phagocytosis assays differ, which may reveal mechanistic subtleties or novel cross-dependencies in shaking cultures or involving other components (*Lima et al., 2014*). It is also suggested that chemotactically active neutrophils are more efficient in blocking the growth of infective fungi, subsequent to phagocytosis (*Jones et al., 2016*).

We have demonstrated a unique ability to detect migration toward bacteria, in a cell number dependent manner and correlated *Dictyostelium* sensitivity for detection of bacterial chemoattractant production. The system was extended to investigate interdependencies of chemotaxis and phagocytosis and can be used to examine virulent and non-virulent bacterial strains, search for novel chemoattractants and their orphan receptors, and screen for molecular inhibitors of chemotaxis/phagocytosis, and further applied to macrophages, neutrophils, as well as *Dictyostelium*.

## Materials and methods

### *Dictyostelium* strains, culturing, and engineering

*Dictyostelium* were grown in HL-5 medium at 22°C in shaking culture, at ~200 rpm, to $1 \times 10^6$ cells/mL. Null strains for *CAR1*, G$\alpha$2, G$\alpha$4, and G$\beta$ were from dictyBase (*Basu et al., 2013*; *Fey et al., 2013*) and grown under selected conditions, as described. Strain phenotypes were confirmed by appropriate response to folate or cAMP in chemotaxis (*Liao et al., 2016*) and biochemical stimulation assays (*Meena and Kimmel, 2016*). The WT CAR1-expressing vector (*Brzostowski et al., 2013*) was transformed into the *car1*-null strain, and CAR1 expression was confirmed by immunoblot with

α-CAR1 (*Klein et al., 1987*). GrlL (*Basu et al., 2013*) was identified in a screen of GPCRs expressed in growing *Dictyostelium* (NPM and ARK, in preparation) and is identical to fAR1 (*Pan et al., 2016*). *grlLlfar1*-null cells were generated by homologous recombination using a targeting construct that had replaced an internal *GrlL/fAR1* gene segment by substitution with a blasticidin resistance cassette (*Faix et al., 2013*). Disruption of the endogenous *GrlL/fAR1* locus was confirmed by PCR amplification, using primers specific to the Blast cassette and an upstream genomic region, and by loss of amplification, using primers specific to the targeted deleted sequence. Full-length GrlL/fAR1 was then re-expressed in the *grlL*-null cells. Full length GrlL/fAR1 cDNA was amplified by RT-PCR from total *Dictyostelium* vegetative RNA sub-cloned into the *Dictyostelium* expression vector pDM353 (*Veltman et al., 2009*), which adds a C-terminal GFP tag. Positive expression was confirmed by sequencing and immunoblotting with α-GFP.

## Chemotaxis of growth-phase *Dictyostelium* to folate and cAMP

*Dictyostelium* were washed twice with 10 mM phosphate buffer, pH 6.5 (PB), and resuspended in PB at $1.5–4.5 \times 10^4$ cells/mL. 60 µL ($1–3 \times 10^3$) growth-phase *Dictyostelium* were added to an upper well of a chemotaxis plate (Essen Bioscience; cat # 4582); in some instances, chemoattractants were included. 200 µL PB (with or without chemoattractants) was added to the bottom well and the plate was transferred to the IncuCyte chamber, at 22°C. Chemotaxis was recorded over time as a function of the cells imaged at the under surface of the membrane, using a 10x objective lens; data were exported, analyzed, and graphed using Microsoft Excel. For each collective experiment, a single end-time point was chosen as a normalizing parameter; often this was a WT control. The exact normalizing control is described separately for each experiment in their respective figure legends.

## Chemotaxis of *Dictyostelium* to bacteria

*K. planticola* and *B. subtilis* [biosafety level 1 (USA)] were grown in LB medium at 37°C, in shaking culture at ~200 rpm, to $OD_{600} <1$. Cells were washed twice with PB and resuspended to the appropriate cell density, with 200 µL bacteria added to the bottom well of a chemotaxis plate. Chemotaxis was otherwise as described above.

## Assay for cAMP production by *K. planticola* and *B. subtilis*

*K. planticola* and *B. subtilis* were grown in LB medium at 37°C, in shaking culture at ~200 rpm, to $OD_{600} <1$. Cells were washed twice with PB and resuspended at different cell densities and incubated at 22°C. Bacterial supernatants were collected at various times, after centrifugation at 12,000 rpm at 4°C, and assayed for accumulated cAMP by ELISA (Cell Biolabs, Inc.; cat # STA-501).

### Phagocytosis assay

Phagocytosis was performed using pHrodo Red *E. coli* BioParticles Conjugate (ThermoFisher Scientific cat#P35361). pHrodo is a fluorogenic dye that exhibits a dramatic increase in fluorescence at an acidified pH pHrodo-labelled bacteria show minimal basal fluorescence [(*Kapellos et al., 2016*) see *Figure 7A*], but upon phagocytosis by *Dictyostelium* and transport to lysosomes, detected fluorescence increases (*Figure 7A*). For some experiments, WT *Dictyostelium* were first stimulated with either buffer control, 50 nM cAMP, or 50 nM folate at 6 min intervals for 1 hr to mimic progressive chemotactic relay-response. The secretion of PDE and folate de-aminase prevents the accumulation of either chemoattractant.

The phagocytosis assay was performed in a 96 well plate. $~0.9 \times 10^5$ *Dictyostelium* in 100 µL PB were added to a select well and allowed to adhere to the well base for 1–5 min. Subsequently, ~10 µg of pHrodo Red *E. coli* BioParticles in 100 µL PB was added to each well. Fluorescent [Excitation/ Emission (nm) 560/585] images were acquired and processed at regular time intervals (~10 to~15 min). Data were exported, analyzed, and graphed using Microsoft Excel.

### Preparation of pHrodo-labelled live *Bacillus subtilis*

*Bacillus subtilis* were grown and harvested as described above. For labelling, $~2 \times 10^9$ bacteria were centrifuged for 1 min at 12,000 rpm and resuspended in ~50 µL of Component C (ThermoFisher Scientific, pHrodo Phagocytosis particle labelling kit; cat # A10026). Succinimidyl ester dye (10 mM) was added to 0.5 mM and incubated for 45 min at room temperature. Toxic compounds (*e.g.*

methanol) were eliminated from the labelling. To remove unlabelled dye, bacteria were washed three times at 12,000 rpm with Component C. pHrodo-labelled *Bacillus subtilis* were viable, as observed by cell division and motility.

For chemotaxis, pHrodo-labelled *Bacillus subtilis* were used at $35 \times 10^6$ cells/well, but was otherwise as described above.

For phagocytosis, pHrodo-labelled *Bacillus subtilis* were used at a 10-fold excess to *Dictyostelium*, but was otherwise as described above.

## Preparation of pHrodo-labelled latex beads

Aliphatic Amine Latex Beads, 1.0 µm (ThermoFisher Scientific; cat. # A37362), were utilized for labeling. ~$5 \times 10^9$ beads were centrifuged for 2 min at 13,000 rpm, washed 3 times with 200 µL of Component C (ThermoFisher Scientific; pHrodo Phagocytosis particle labeling kit, cat. # A10026), and resuspended in 50 µL of Component C. Succinimidyl ester (pHrodo Red) dye (10 mM) was added to 0.5 mM (at <1% the total number of free amino groups in the bead population) and incubated for 45 min at room temperature. Unlabelled dye was removed by washing the beads three times with Component C. The beads were resuspended in PB at $5 \times 10^6$ beads/µL.

For phagocytosis, ~$5 \times 10^8$ pHrodo-labeled beads and ~$1 \times 10^5$ *Dictyostelium* were used per well, but was otherwise as described above.

## Folate-PEG-FITC labelling of pHrodo-labelled latex beads

To evaluate the effect of folate at the bead surface for phagocytosis, we covalently coupled a chemo-active folate derivative to the free amine groups ($>10^6$ sites/bead) at the surface of pHrodo-labelled latex beads, using 1-Ethyl-3-(3-dimethylaminopropyl) carbodiimide hydrochloride (Thermo-Fisher Scientific; EDAC, cat. # E2247) as a linker. Briefly, 0.5 mM Folate-PEG-FITC (Nanocs; cat. # PG2-FAFC-2k) and 2 mM EDAC were incubated with 200 µL of pHrodo-labelled latex beads for 16 hr at 4°C. Beads were washed three times with PB at 13,000 rpm for 2 min and finally suspended in PB at $5 \times 10^6$ beads/µL.

Bound Folate-PEG-FITC was quantified by measuring the fluorescence of FITC ($\lambda ex$ 488 nm; $\lambda em$ 525 nm) in a microplate reader (Spectra MAX Gemini EM, Molecular Devices). We estimate there are an average of 20,000 folate sites per bead (d = 1.0 µm).

Correcting for calculated size-dependent differences in rates of diffusion, folate-PEG-FITC shows a similar $EC_{50}$ for chemotaxis of WT *Dictyostelium* as for folate. We also demonstrate that Folate-PEG-FITC/pHrodo-labelled latex beads can elicit a biochemical response in a stimulation assay for folate.

For phagocytosis, ~$5 \times 10^8$ pHrodo-/Folate-PEG-FITC-labelled beads and ~$10^5$ *Dictyostelium* were used per well, but was otherwise as described above.

## Immunofluorescence microscopy of pHrodo-labeled and Folate-PEG-FITC/pHrodo-labeled latex beads

To assess uniformity of Folate-PEG-FITC and pHrodo labelling at bead surfaces and concordance of both Folate-PEG-FITC and pHrodo labels on individual beads, we visualized bead fluorescence by confocal microscopy (see *Figure 9—figure supplements 1* and *2*). Briefly, pHrodo-labelled and Folate-PEG-FITC/pHrodo-labelled latex beads were placed in Lab-Tec chamber slides (Thermo Fisher Scientific) and confocal images (LSM 780 confocal/multiphoton microscope, Carl Zeiss, Thornwood, NY, USA) were acquired for DIC, rhodamine, and FITC and processed using Zeiss ZEN microscope software.

## pPKBR1 and pERK assays

The phosphorylation of ERK2 or PKBR1 in growth-phase cells after stimulation with cAMP, folate, or bacterial supernatants was as previously described (*Meena and Kimmel, 2016*), with slight modifications. Briefly, cells were grown in shaking culture, at 200 rpm, to ~$1–2 \times 10^6$ cells/mL at 22°C. Cells were re-plated at ~$3 \times 10^6$ cells/well in six well plates and allowed to adhere. After ~1 hr, cells were washed and overlayered with 1 mL of PB. Cells were stimulated and after various times, responses were terminated by addition of 200 µL of 1x Laemmli sample buffer, with 10% $\beta$ME. Samples were incubated at 95°C for 10 min and 15 µL of cell lysate was utilized for immunoblotting for pERK2,

pPKBR1, and Actin, as described (*Meena and Kimmel, 2016*). Full time course (0, 15, 30, 60, 180 s) assays for activation and adaptation were monitored for each experiment. For comparative activations, data are shown for 15 s maxima for pPKBR1 and 30 s maxima for pERK2 (*Meena and Kimmel, 2016*).

### EZ-TaxiScan chemotaxis of growth-phase *Dictyostelium*

Chemotaxis assays were carried out using the EZ-TAXIScan chamber (Effector Cell Institute, Tokyo, Japan) as previously described (*Liao et al., 2016*), with slight minor modifications. Briefly, cells were grown in shaking culture, at 200 rpm, to ~1 × 10⁶ cells/mL at 22°C. These growing *Dictyostelium* were washed twice with PB, and ~2000 cells were loaded into the starting well and buffer was drawn from the opposing well to cause flow. Once more than 100 cells accumulated along the starting border of the terrace, flow was stopped by reintroducing buffer into the opposing well. 10 μM of buffer, cAMP, or folate was added to the opposing well. Migration was recorded with 30 s intervals for 120 min at 22°C.

## Acknowledgements

This work was supported by the Intramural Research Program of the National Institute of Diabetes and Digestive and Kidney Diseases, National Institutes of Health. We thank colleagues who have developed cell lines and, especially, dictyBase (http://dictybase.org/).

## Additional information

### Funding

| Funder | Author |
| --- | --- |
| NIH Office of the Director | Netra Pal Meena |

The funders had no role in study design, data collection and interpretation, or the decision to submit the work for publication.

### Author contributions

NPM, Conceptualization, Data curation, Software, Formal analysis, Validation, Investigation, Visualization, Methodology, Writing—original draft, Writing—review and editing; ARK, Conceptualization, Resources, Data curation, Software, Formal analysis, Supervision, Funding acquisition, Validation, Investigation, Visualization, Methodology, Writing—original draft, Project administration, Writing—review and editing

### Author ORCIDs

Alan R Kimmel, http://orcid.org/0000-0002-0533-1939

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
