## [Decision Letter]

Thank you for submitting your article "Chemotaxis to Live Bacteria, Network Responses, and Independence of Phagocytosis from Chemo-Receptor Sensing" for consideration by *eLife*. Your article has been favorably evaluated by Naama Barkai (Senior Editor) and three reviewers, one of whom is a member of our Board of Reviewing Editors. The reviewers have opted to remain anonymous.

The reviewers have discussed the reviews with one another and the Reviewing Editor has drafted this decision to help you prepare a revised submission.

The reviewers were impressed with the technology and the experimental design and all of them thought that publication of this paper would constitute a significant contribution to the field, but they also expressed concerns with several aspects of the manuscript that preclude acceptance in the present form. Reviewers #2 and #3 expressed major reservations regarding the relationship between phagocytosis and chemotaxis. Reviewer #2 pointed out that human neutrophils kill fungi more efficiently after chemotaxis and suggested that probing this question in *Dictyostelium* would make the manuscript more suitable for publication. Reviewer #3 noted that publications from the 1980s already showed that chemotaxis and phagocytosis were independent processes because *Dictyostelium* cells can phagocytize dead bacteria and even latex beads. Upon further discussion, the reviewers agreed that the authors should test directly whether phagocytosis is indeed affected by chemotaxis in *Dictyostelium* cells. They also agreed that the issue of phagocytosis of inanimate objects should be addressed explicitly in the manuscript. The authors may also want to test whether phagocytosis of inanimate objects, such as latex beads, is affected by chemotaxis, but they should definitely mention this fact and discuss it in the context of their findings.

Overall, these experiments may change the general conclusions of the manuscript so another round of review would be necessary.

Reviewer #1:

This is an excellent manuscript that describes an important biological phenomenon – chemotaxis of *D. discoideum* amoebae to bacteria. It also explores the relationship between chemotaxis and phagocytosis. The manuscript provides convincing evidence that the amoebae utilize several sensing mechanisms to detect and approach bacteria by chemotaxis and that phagocytosis is largely distinct from chemotaxis. The experimental design is clever and careful, the results are clear and believable, and the authors have adopted several new methods to the system, which will have a significant impact on future investigations. Most importantly, the conclusions are largely supported by the data and the discussion is thorough and interesting. Overall this manuscript is very strong and it does not suffer from major concerns.

Reviewer #2:

The manuscript from Meena and Kimmel presents a series of interesting observations regarding the chemotaxis of *Dictyostelium* towards bacteria. Several interesting insights emerge from these observations: (1) the migration of *Dictyostelium* cells towards bacteria is guided by multiple chemoattractants. The role of GPCRs in migration guided by folate is well characterized. (2) The adaptation to various chemoattractants uses specific adaptation mechanisms that are distinct for each chemoattractant. This is important because *Dictyostelium* could remain sensitive to one chemoattractant while at saturating concentrations of another one. (3) phagocytosis and chemotaxis are independent processes.

The relationship between phagocytosis and chemotaxis presented in the manuscript is weak. The major question is not necessarily if phagocytosis and chemotaxis are independent processes. Most important is what is the end-result of the interaction? While the phagocytosis and chemotaxis appear to be independent, the killing ability of *Dictyostelium* may be enhanced after chemotaxis. Recently, it was reported that human neutrophils kill fungi more efficiently after chemotaxis compared to exposure to a uniform concentration of chemoattractant (no chemotaxis – Jones et al., JID 2015). Could it be true that *Dictyostelium* is a more efficient killer after chemotaxis – supporting a relationship between chemotaxis and phagocytosis. Probing the killing of bacteria strains that are more difficult to kill than the two tested so far may reveal some interesting answers?

Reviewer #3:

The authors have carried out a number of well-controlled experiments to tease out the relative contributions of folate and/or cAMP chemotaxis by *Dictyostelium* amoebae in bacterial phagocytosis. They have done an excellent job distinguishing these two actin-dependent processes from one another and conclude that bacterial phagocytosis is independent of chemotaxis to either folate or cAMP. There is much else to admire in the experiments they present. For example, their demonstration that the adaptation pathways for folate and cAMP function independently of one another and upstream of common downstream cellular pathways is particularly convincing. For this non-cell biologist, this is one of the clearest demonstrations of this concept.

The authors appropriately focus on the last 5-10 years of others' work on the topic because there has been much progress in deciphering the molecular mechanisms of actin-dependent processes during this time, but I was left wondering why there is no comment on the work done in the last century. We have known for decades that *Dictyostelium* phagocytoses autoclaved bacteria (that are presumably devoid of chemoattractants) and 1-µm latex beads! For the later, there was a pretty clear demonstration of altered binding to latex beads in phagocytosis mutants:

Vogel, G., L. Thilo, H. Schwarz and R. Steinhart (1980). "Mechanism of phagocytosis in *Dictyostelium discoideum*: Phagocytosis is mediated by different recognition sites as disclosed by mutants with altered phagocytotic properties." J. Cell Biol. 86: 456-465.

I do not wish to detract from the excellent experiments here, or be overly critical, since I do feel this work advances the field and provides new and convincing information, but I have always felt that the phagocytosis of large latex beads demonstrates that phagocytosis is independent of chemotactic mechanisms. Since this is one of the major claims that the authors are making in this work, they may wish to discuss the phagocytosis of inanimate objects published previously.

---

## [Author Response]

The reviewers were impressed with the technology and the experimental design and all of them thought that publication of this paper would constitute a significant contribution to the field, but they also expressed concerns with several aspects of the manuscript that preclude acceptance in the present form. Reviewers #2 and #3 expressed major reservations regarding the relationship between phagocytosis and chemotaxis. Reviewer #2 pointed out that human neutrophils kill fungi more efficiently after chemotaxis and suggested that probing this question in Dictyostelium would make the manuscript more suitable for publication. Reviewer #3 noted that publications from the 1980s already showed that chemotaxis and phagocytosis were independent processes because Dictyostelium cells can phagocytize dead bacteria and even latex beads. Upon further discussion, the reviewers agreed that the authors should test directly whether phagocytosis is indeed affected by chemotaxis in Dictyostelium cells. They also agreed that the issue of phagocytosis of inanimate objects should be addressed explicitly in the manuscript. The authors may also want to test whether phagocytosis of inanimate objects, such as latex beads, is affected by chemotaxis, but they should definitely mention this fact and discuss it in the context of their findings.

Although it is well known that *Dictyostelium* will phagocytose latex beads without attached chemoattractant ligands, this was never considered sufficient proof that chemo-sensing does not influence phagocytosis. Comparative testing of e.g. folate-coated latex beads with un-coated controls was never studied in the previous manuscript. We have now added additional data to this end. We show that rates of phagocytosis with control and folate-beads are identical (see Figure 9), further supporting our previous conclusions that phagocytosis is not modulated by chemo-receptor sensing. We have also experimentally addressed the question if chemotactically active cells have increased phagocytic rates compared to quiescent cells (see Figure 8). Chemotactically active and quiescent cells have similar phagocytic rates. These new data also support our previous conclusions.

Overall, these experiments may change the general conclusions of the manuscript so another round of review would be necessary.

The experimental suggestions were very welcome. All the new data are in perfect agreement with our original conclusions.

Reviewer #2:

The manuscript from Meena and Kimmel presents a series of interesting observations regarding the chemotaxis of Dictyostelium towards bacteria. Several interesting insights emerge from these observations: (1) the migration of Dictyostelium cells towards bacteria is guided by multiple chemoattractants. The role of GPCRs in migration guided by folate is well characterized. (2) The adaptation to various chemoattractants uses specific adaptation mechanisms that are distinct for each chemoattractant. This is important because Dictyostelium could remain sensitive to one chemoattractant while at saturating concentrations of another one. (3) phagocytosis and chemotaxis are independent processes.

Thank you for the careful reading and commentary. The summary for points #1 and #2 is accurate, but we must not have been sufficiently clear to other data for the referee to conclude for #3 that we made a definitive statement with “phagocytosis and chemotaxis being independent processes”. We present very strong data that “Phagocytosis is not dependent upon Chemo-Receptor Sensing”. That is, activation of chemoattractant receptors by secreted or surface immobilized ligands is neither activating nor inhibitory to phagocytosis. With increased and directed motility, chemotaxing cells will encounter and engulf “prey” more efficiently than non-chemotactic cells. However, our data are not consistent with phagocytic efficiency being modulated by pathways that act downstream of chemoattractant receptor activation.

In an effort to address this further, we compared phagocytic efficiency of WT cells that had or had not been exposed to pulsatory stimulation of either folate or cAMP chemoattractants. These conditions create periodic chemoattractant activation/adaptation cycling at 6 min intervals, which mimics chemoattractant wave propagation and response (see new Figure 8). Chemoattractant activation did not stimulate phagocytosis relative to unstimulated controls.

We have also extended the question further by comparing phagocytic rates of WT cells for 1 μm latex beads with or without folate surface coats (see new Figure 9), or for WT and folate receptor-null cells for folate-coated beads. No differences in rates of phagocytosis were observed among the different experiments and controls.

The relationship between phagocytosis and chemotaxis presented in the manuscript is weak. The major question is not necessarily if phagocytosis and chemotaxis are independent processes. Most important is what is the end-result of the interaction? While the phagocytosis and chemotaxis appear to be independent, the killing ability of Dictyostelium may be enhanced after chemotaxis. Recently, it was reported that human neutrophils kill fungi more efficiently after chemotaxis compared to exposure to a uniform concentration of chemoattractant (no chemotaxis – Jones et al., JID 2015). Could it be true that Dictyostelium is a more efficient killer after chemotaxis – supporting a relationship between chemotaxis and phagocytosis. Probing the killing of bacteria strains that are more difficult to kill than the two tested so far may reveal some interesting answers?

We recognize the referee’s focus to the multifaceted connections of chemotaxis and phagocytosis. We had tried to be very specific to examine if “Chemo-Receptor Sensing” influenced phagocytosis. As mentioned above (see new Figure 8, Figure 9), we have added significant new supporting data. We believe that our data strongly indicate that phagocytosis is not dependent upon chemo-receptor sensing. The comparative “killing” ability of chemotaxing or non-chemotaxing cells is quite a separate and very distinct question, from phagocytosis per se. Indeed, the assay with essential control normalizations would require new developments, which we feel are beyond the important questions that we have addressed directly. Nonetheless, we have added the concept to the Discussion with appropriate reference. We hope the referee is satisfied with our explanation and with the new data.

Reviewer #3:

[…] The authors appropriately focus on the last 5-10 years of others' work on the topic because there has been much progress in deciphering the molecular mechanisms of actin-dependent processes during this time, but I was left wondering why there is no comment on the work done in the last century. We have known for decades that Dictyostelium phagocytoses autoclaved bacteria (that are presumably devoid of chemoattractants) and 1-µm latex beads! For the later, there was a pretty clear demonstration of altered binding to latex beads in phagocytosis mutants:

Vogel, G., L. Thilo, H. Schwarz and R. Steinhart (1980). "Mechanism of phagocytosis in Dictyostelium discoideum: Phagocytosis is mediated by different recognition sites as disclosed by mutants with altered phagocytotic properties." J. Cell Biol. 86: 456-465.

I do not wish to detract from the excellent experiments here, or be overly critical, since I do feel this work advances the field and provides new and convincing information, but I have always felt that the phagocytosis of large latex beads demonstrates that phagocytosis is independent of chemotactic mechanisms. Since this is one of the major claims that the authors are making in this work, they may wish to discuss the phagocytosis of inanimate objects published previously.

Rather than just discuss this, we decided to address the question by direct experimentation. While one may argue that the ability of *Dictyostelium* to engulf latex beads argues that chemotactic response has no function in phagocytosis, this question had not been addressed formally in the previous manuscript. Here, we generate folate-coated latex beads and directly compare phagocytosis to that of non-folate controls (see Figure 9), for WT and folate receptor-null cells. No differences were observed supporting our conclusions that chemo-sensing does not have a significant role in phagocytosis.